## META-RESEARCH

# Lessons from a catalogue of 6674 brain recordings

**Abstract**  It is now possible for scientists to publicly catalogue all the data they have ever collected on one phenomenon. For a decade, we have been measuring a brain response to visual symmetry called the sustained posterior negativity (SPN). Here we report how we have made a total of 6674 individual SPNs from 2215 participants publicly available, along with data extraction and visualization tools (https://osf.io/2sncj/). We also report how re-analysis of the SPN catalogue has shed light on aspects of the scientific process, such as statistical power and publication bias, and revealed new scientific insights.

**ALEXIS DJ MAKIN\*, JOHN TYSON-CARR, GIULIA RAMPONE, YIOVANNA DERPSCH, DAMIEN WRIGHT AND MARCO BERTAMINI**

## Introduction

Many natural and man-made objects are symmetrical, and humans can detect visual symmetry very efficiently (*Bertamini et al., 2018*; *Treder, 2010*; *Tyler, 1995*; *Wagemans, 1995*). Visual symmetry has been a topic of research within experimental psychology for more than a century. In recent decades, two techniques – functional magnetic resonance imaging (fMRI) and electroencephalography (EEG) – have been used to investigate the impact of visual symmetry on a region of the brain called the extrastriate cortex. Since 2011, we have been using EEG in experiments at the University of Liverpool to measure a brain response to visual symmetry called the sustained posterior negativity (SPN; see *Box 1* and *Figure 1*). By October 2020 we had completed 40 SPN projects: 17 of these had been published, and the remaining 23 were either unpublished or under review.

The COVID pandemic stopped all our EEG testing in March 2020, and we used this crisis/opportunity to organize and catalogue our existing data. The data from all 40 of our SPN projects are now available in a public repository called "The complete Liverpool SPN catalogue" (available at https://osf.io/2sncj/; see *Box 2* and *Figure 2*). The catalogue allows us to draw conclusions that could not be gleaned from a single experiment. It can also support meta-scientific evaluation of our data and practices, as reported in the current article.

## Meta-scientific lessons from the complete Liverpool SPN catalogue

There is growing anxiety about the trustworthiness of published science (*Munafò et al., 2017*; *Open Science Collaboration, 2015*; *Errington et al., 2021*). Many have argued that we should build cumulative research programs, where effects are measured reliably, and the truth becomes clearer over time. However, common practice often falls far short of this ideal. And although there has been a positive response to the replication crisis in psychology (*Nelson et al., 2018*), there is still – according to the cognitive neuroscientist Dorothy Bishop – room for improvement: "many researchers persist in working in a way almost guaranteed not to deliver meaningful results. They ride what I refer to as the four horsemen of the irreproducibility apocalypse" (*Bishop, 2019*). The four horsemen are: (i) publication bias; (ii) low statistical power; (iii) p value hacking; (iv) HARKing (hypothesizing after results known).

The "manifesto for reproducible science" includes these four items and two more: poor quality control in data collection and analysis, and the failure to control for bias (*Munafò et al., 2017*). Such critiques challenge all scientists to answer a simple question: are you practicing cumulative science, or is your research is undermined by the four horsemen of the irreproducibility apocalypse? Indeed, before compiling the

**\*For correspondence:**
alexis.makin@liverpool.ac.uk

**Competing interest:** The authors declare that no competing interests exist.

## Box 1. Symmetry and the sustained posterior negativity (SPN).

Visual symmetry plays an important role in perceptual organization (*Koffka, 1935*; *Wagemans et al., 2012*) and mate choice (*Grammer et al., 2003*). This suggests sensitivity to visual symmetry is innate: however, symmetrical prototypes could also be learned from many asymmetrical exemplars (*Enquist and Johnstone, 1997*). Psychophysical experiments have taught us a great deal about symmetry perception (*Barlow and Reeves, 1979*; *Treder, 2010*; *Wagemans, 1995*), and the neural response to symmetry has been studied more recently (for reviews see *Bertamini and Makin, 2014*; *Bertamini et al., 2018*; *Cattaneo, 2017*). Functional MRI has reliably found symmetry activations in the extrastriate visual cortex (*Chen et al., 2007*; *Keefe et al., 2018*; *Kohler et al., 2016*; *Sasaki et al., 2005*; *Tyler et al., 2005*; *Van Meel et al., 2019*). The extrastriate symmetry response can also be measured with EEG. Visual symmetry generates an event related potential (ERP) called the sustained posterior negativity (SPN). The SPN is a difference wave – amplitude is more negative at posterior electrodes when participants view symmetrical displays compared to asymmetrical displays (*Jacobsen and Höfel, 2003*; *Makin et al., 2012*; *Makin et al., 2016*; *Norcia et al., 2002*). As shown in *Figure 1*, SPN amplitude scales parametrically with the proportion of symmetry in the image (*Makin et al., 2020c*).

SPN catalogue, we were unable to answer this question for our own research program.

One problem with replication attempts is their potentially adversarial nature. Claiming that other people's published effects are unreliable insinuates bad practice, while solutions such as "adversarial collaboration" are still rare (*Cowan et al., 2020*). In this context and heeding the call to make research in psychology auditable (*Nelson et al., 2018*), we decided to take an exhaustive and critical look at the complete SPN catalogue in terms of the four horsemen of irreproducibility.

### Horseman one: publication bias

Most scientists are familiar with the phrase "publish or perish" and know it is easier to publish a statistically significant effect ($P<.05$). Null results accumulate in the proverbial file drawer, while false positives enter the literature. This publication bias leads to systematic overestimation of effect sizes in meta-analysis (*Brysbaert, 2019*; *Button et al., 2013*).

The cumulative distribution of the 249 SPN amplitudes is shown in *Figure 3A* (top panel), along with the cumulative distributions for those in the literature and those in the file drawer (middle panel). The unpublished SPNs were weaker than the published SPNs (mean difference = 0.354 microvolts [95% CI=0.162–0.546], t (218.003)=3.640, $P<.001$, equal variance not assumed). Furthermore, the published SPNs came from experiments with smaller sample sizes

(mean sample sizes = 23.40 vs 29.49, $P<.001$, Mann-Whitney U test).

To further explore these effects, we ran three meta-analyses (using the metamean function from the dmetar library in R). Full results are described in supplementary materials (https://osf.io/q4jfw/). The weighted mean amplitude of the published SPNs was –1.138 microvolts [95% CI = –1.290; –0.986]. This was reduced to –0.801 microvolts [–0.914; –0.689] for the unpublished SPNs, and to –0.954 microvolts [–1.049; –0.860] for all SPNs. The funnel plot for the published SPNs (*Figure 3A*; bottom panel) is not symmetrically tapered (less accurate measures near the base of the funnel are skewed leftwards). This is a textbook fingerprint of publication bias. However, the funnel asymmetry was still significant for the unpublished SPNs and for all SPNs, so publication bias cannot be the explanation (see *Zwetsloot et al., 2017* for detailed analysis of funnel asymmetry).

The amplitudes of the P1 peak and the N1 trough from the same trials provide an instructive comparison. Our SPN papers do not need large P1 and N1 components, so these are unlikely to have a systematic effect on publication. P1 peak was essentially identical in published and unpublished work (4.672 vs 4.686, t (195.11) = –0.067, $P=.946$; *Figure 3B*). This is potentially an interesting counterpoint to the SPN. However, the N1 trough was larger in published work (–8.736 vs. –7.155. t (183.61) = –5.636, $P<.001$; *Figure 3C*).

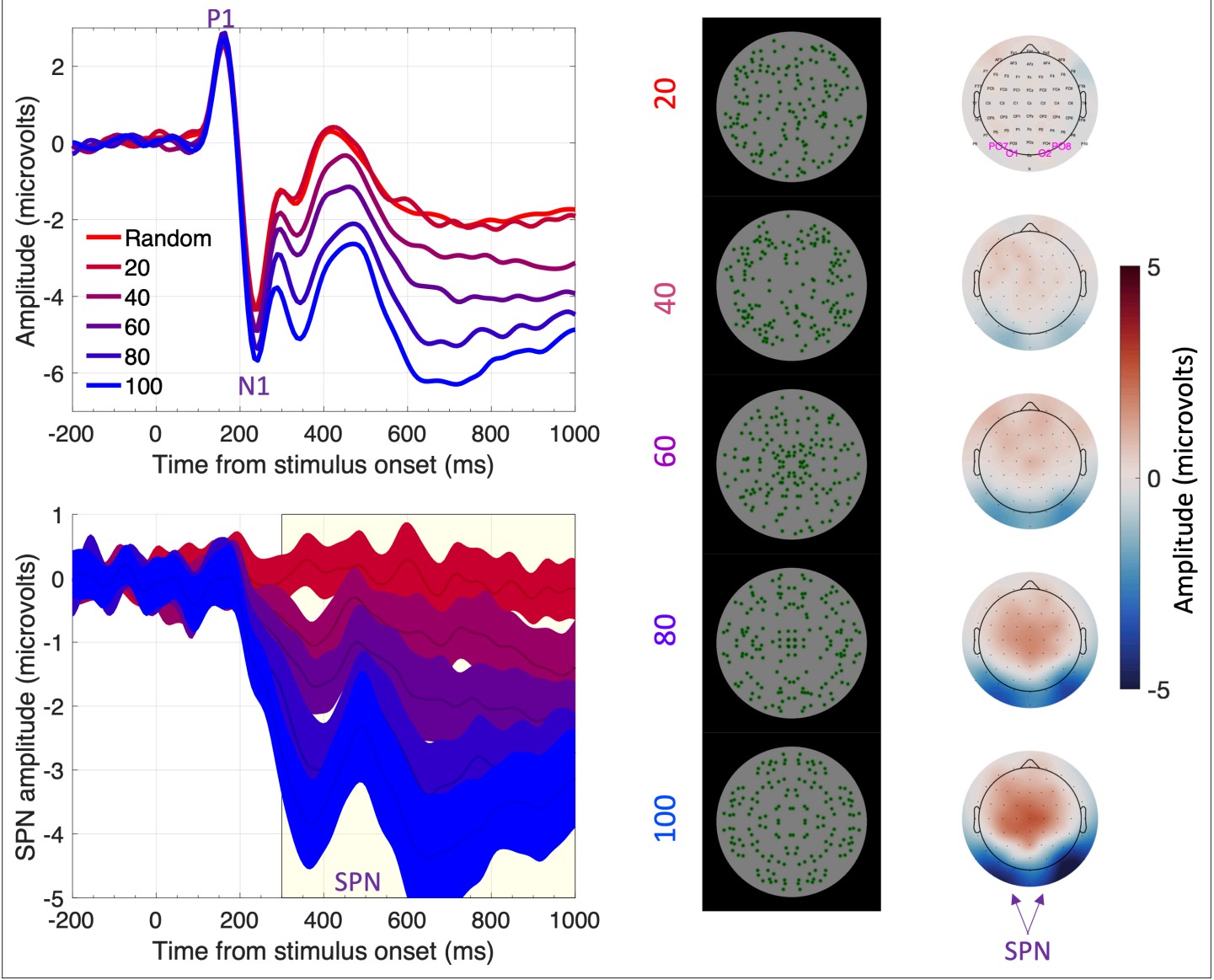

**Figure 1.** The sustained posterior negativity (SPN). The grand-average ERPs are shown in the upper left panel and difference waves (reflection-random) are shown in the lower left panel. A large SPN is a difference wave that falls a long way below zero. Topographic difference maps are shown on the right, aligned with the representative stimuli (black background). The difference maps depict a head from above, and the SPN appears as blue at the back. Purple labels indicate electrodes used for ERP waves [PO7, O1, O2 and PO8]. Note that SPN amplitude increases (that is, becomes more negative) with the proportion of symmetry in the image. In this experiment, the SPN increased from ~0 to –3.5 microvolts as symmetry increased from 20% to 100%. Adapted from Figures 1, 3 and 4 in *Makin et al., 2020c*.

The funnel asymmetry was also significant for both P1 and N1.

### Summary for horseman one: publication bias

This analysis suggests a tendency for large SPNs to make it into the literature, and small ones to linger in the file drawer. In contrast, the P1 was essentially identical in published and unpublished work. However, we do not think publication bias is a problem in our SPN research.

The published and unpublished SPNs are from heterogenous studies, with different stimuli and tasks. We would not necessarily expect them to have the same mean amplitude. In other words, the SPN varies across studies not only due to neural noise and measurement noise, but also due to experimental manipulations affecting the neural signal (although W-load and Task were similar in published and unpublished work, see *Box 2*). Furthermore, it is *not* the case that our published papers selectively report one side of a

## Box 2. The complete Liverpool SPN catalogue.

The complete Liverpool SPN catalogue was compiled from 40 projects, each of which involved between 1 and 5 experiments (with an experiment being defined as a study producing a dataset composed only of within-subject conditions). Sample size ranged from 12 to 48 participants per experiment (mean = 26.37; mode = 24; median = 24). Each experiment provided 1–8 grand-average SPN waves. In total, we reanalysed 249 grand-average SPNs, 6,674 participant-level SPNs, and 850,312 single trials. SPNs from other labs are not yet included in the catalogue (*Höfel and Jacobsen, 2007a*; *Höfel and Jacobsen, 2007b*; *Jacobsen et al., 2018*; *Kohler et al., 2018*; *Martinovic et al., 2018*; *Wright et al., 2018*). Steady-state visual evoked potential responses to symmetry are also unavailable (*Kohler et al., 2016*; *Kohler and Clarke, 2021*; *Norcia et al., 2002*; *Oka et al., 2007*). However, the intention is to keep the catalogue open, and the design allows many contributions. In the future we hope to integrate data from other labs. This will increase the generalizability of our conclusions. Anyone wishing to use the catalogue can start with the beginner's guide, available on open science framework (https://osf.io/bq9ka/).

The catalogue also includes several supplementary files, including a file called "One SPN Gallery.pdf" (https://osf.io/eqhd5/) which has one page for each of the 249 SPNs, along with all technical information about the stimuli and analysis (*Figure 2* shows the first SPN from the gallery). Browsing this gallery reveals that 39/40 projects used abstract stimuli, such as dot patterns or polygons (see, for example, Project 1: *Makin et al., 2012*). The exception was Project 14, which used flowers and landscapes (*Makin et al., 2020b*). The SPN is generated automatically when symmetry is present in the image (e.g., Project 13: *Makin et al., 2020c*). However, the brain can sometimes go beyond the image and recover symmetry in objects, irrespective of changes in view angle (Project 7: *Makin et al., 2015*).

Almost half the SPNs (125/249) were recorded in experiments where participants were engaged in active regularity discrimination (e.g., press one key to report symmetry and another to report random). The other 124 SPNs were recorded in conditions where participants were performing a different task, such as discriminating the colour of the dots or holding information in visual working memory (*Derpsch et al., 2021*). In most projects the stimuli were presented for at least 1 second and the judgment was entered in a non-speeded fashion after stimulus offset. Key mapping was usually shown on the response screen to avoid lateralized preparatory motor responses during stimulus presentation.

The catalogue is designed to be FAIR (Findable, Accessible, Interoperable and Reusable). For each project we have included uniform data files from five subsequent stages of the pipeline: (i) raw BDF files; (ii) epoched data before ICA pruning; (iii) epoched data after ICA pruning; (iv) epoched data after ICA pruning and trial rejection; (v) pre-processed data averaged across trials for each participant and condition (stage v is the starting point for most ERP visualization and meta-analysis in this article). The catalogue also includes Brain Imaging Data Structure (BIDS) formatted files from stage iv (https://osf.io/e8r95/). BIDS files from earlier processing stages can be compiled from available codes or GUI (https://github.com/JohnTyCa/The-SPN-Catalogue) by users of MATLAB with EEGLAB and BIOSEMI toolbox.

Furthermore, we developed an app that allows users to: (a) view the data and summary statistics as they were originally published; (b) select data subsets, electrode clusters, and time windows; (c) visualize the patterns; (d) export data for further statistical analysis. This is available to Windows or Mac users with a Matlab license, and a standalone version can be used on Windows without a Matlab license. The app, executable and standalone scripts, and dependencies are available on Github (https://github.com/JohnTyCa/The-SPN-Catalogue, copy archived at swh:1:rev:75e729f867c275433b68807bc3f2228c57a3ccac, *Tyson-Carr, 2022*). This repository and app will be maintained and expanded to accommodate data from future projects.

The folder called "SPN user guides and summary analysis" (https://osf.io/gjpr7/) also contains supplementary files that give all the technical details required for reproducible EEG research, as recommended by the Organization for Human Brain Mapping *Pernet et al., 2020*. For instance, the file called "SPN effect size and power V8.xlsx" has one worksheet for each project (https://osf.io/c8jgy/). This file documents all extracted ERP data along with details about the electrodes, time windows, ICA components removed, and trials removed. With a few minor exceptions, anyone can now reproduce any figure or analysis in our SPN research. Users can also run alternative analyses that depart from the original pipeline at any given stage. Finally, the folder called "Analysis in eLife paper" contains all materials from this manuscript (https://osf.io/4cs2p/).

Although this paper focuses on meta-science, we can briefly summarize the scientific utility of the catalogue. Analysis of the whole data set shows that SPN amplitude scales with the salience of visual regularity. This can be estimated with the 'W-load' from theoretical models of perceptual goodness (*van der Helm and Leeuwenberg, 1996*). SPN amplitude also increases when regularity is task relevant. Linear regression with two predictors (W-load and Task, both coded on a 0–1 scale) explained 33% variance in grand-average SPN amplitude (SPN (microvolts) = –1.669 W – 0.416Task +0.071). The SPN is slightly stronger over the right hemisphere, but the laws of perceptual organization, that determine SPN amplitude, are similar on both sides of the brain. Source dipole analysis can also be applied to the whole data set (following findings of *Tyson-Carr et al., 2021*). We envisage that most future papers will begin with meta-analysis of the SPN catalogue, before reporting a new purpose-built experiment. The SPN catalogue also allows meta-analysis of other ERPs, such as P1 or N1, which may be systematically influenced by stimulus properties (although apparently not W-load).

distribution with a mean close to zero. Our best estimate of mean SPN amplitude (–0.954 microvolts) is far below zero [95% CI = –1.049; –0.860]. The file drawer has no embarrassing preponderance of sustained posterior *positivity*.

Some theoretically important effects can appear robust in meta-analysis of published studies, but then disappear once the file drawer studies are incorporated. Fortunately, this does not apply to the SPN. We suggest that assessment of publication bias is a feasible first step for other researchers undertaking a catalogue-evaluate exercise.

## Horseman two: low statistical power

According to *Brysbaert, 2019*, many cognitive psychologists have overly sunny intuitions about power analysis and fail to understand it properly. The first misunderstanding is that an effect on the cusp of significance (*P*=.05) has a 95% chance of successful replication, when in fact the probability of successful replication is only 50% (power = 0.5). Researchers often work on the cusp of significance, where power is barely more than 0.5. Indeed, one influential analysis estimated that median statistical power in cognitive

neuroscience is just 0.21 (*Button et al., 2013*). In stark contrast, the conventional threshold for adequate power is 0.8. Although things may be improving, many labs still conduct underpowered experiments without sufficient awareness of the problem.

To estimate statistical power, one needs a reliable estimate of effect size, and this is rarely available a priori. In the words of *Brysbaert, 2019*, "you need an estimate of effect size to get started, and it is very difficult to get a useful estimate". It is well known that effect size estimates from published work will be exaggerated because of the file drawer problem (as described previously). It is less well known that one pilot experiment does not provide a reliable estimate of effect size (especially when the pilot itself has a small sample; *Albers and Lakens, 2018*). Fortunately, we can estimate SPN effect size from many experiments, published and unpublished, and this allows informative power analysis.

For a single SPN, the relevant effect size metric is Cohen's $d_z$ (mean amplitude difference/SD of amplitude differences). *Figure 4A* shows the relationship between SPN amplitude and effect size $d_z$. The larger (more negative) the SPN in microvolts, the larger $d_z$. The curve tails off for strong SPNs, resulting in a nonlinear relationship. The second

order polynomial trendline was a better fit than the first order linear trendline (see the supplementary polynomial regression analysis at https://osf.io/f2659/). The same relationship is found whether regularity is task relevant or not (*Figure 4B*) and in published and unpublished work (*Figure 4C*). Crucially, we can now estimate typical effect size for an SPN with a given amplitude using this polynomial regression equation (see the SPN effect size calculator at https://osf.io/gm734/).

This approach can be illustrated with 0.5 microvolt SPNs. Although these are at the low end of the distribution (*Figure 4A*), they can be interpreted and published (e.g., *Makin et al., 2020a*). The average $d_z$ for a 0.5 microvolt SPN is –0.469. Power analysis shows that to have an 80% chance of finding an effect of this size (*P*<.05, two tailed) we need a sample of 38 participants. In contrast, our median sample size is 24, which gives us an observed power of just 60%. In other words, if we were to choose a significant 0.5 microvolt SPN and rerun the exact same experiment, there is a 100%–60%=40% chance we would not find the significant SPN again. This is not a solid foundation for cumulative research.

Only a third of the 249 SPNs are 0.5 microvolts or less. However, many papers do not merely report the presence of a significant SPN. Instead, the headline effect is usually a within-subjects difference between experimental conditions. As a first approximation, we can assume the same power analysis applies to pairwise SPN modulations. We thus need 38 participants for an 80% chance of detecting an ~0.5 microvolt SPN difference between two regular conditions (and more participants for between-subject designs).

The table in *Figure 4G* gives required sample size (N) for 80% chance of obtaining SPNs of a particular amplitude (power = 0.8, alpha = 0.05, two-tailed). This suggests relatively large 1.5 microvolt SPNs could be obtained with just 9 participants. However, estimates of effect size are less precise at the high end (see the supplementary polynomial regression analysis at https://osf.io/f2659). A conservative yet feasible approach is to collect at least 20 participants even when confident of obtaining a large SPN or SPN modulation. Alternatively, researchers may require a sample that allows them to find the

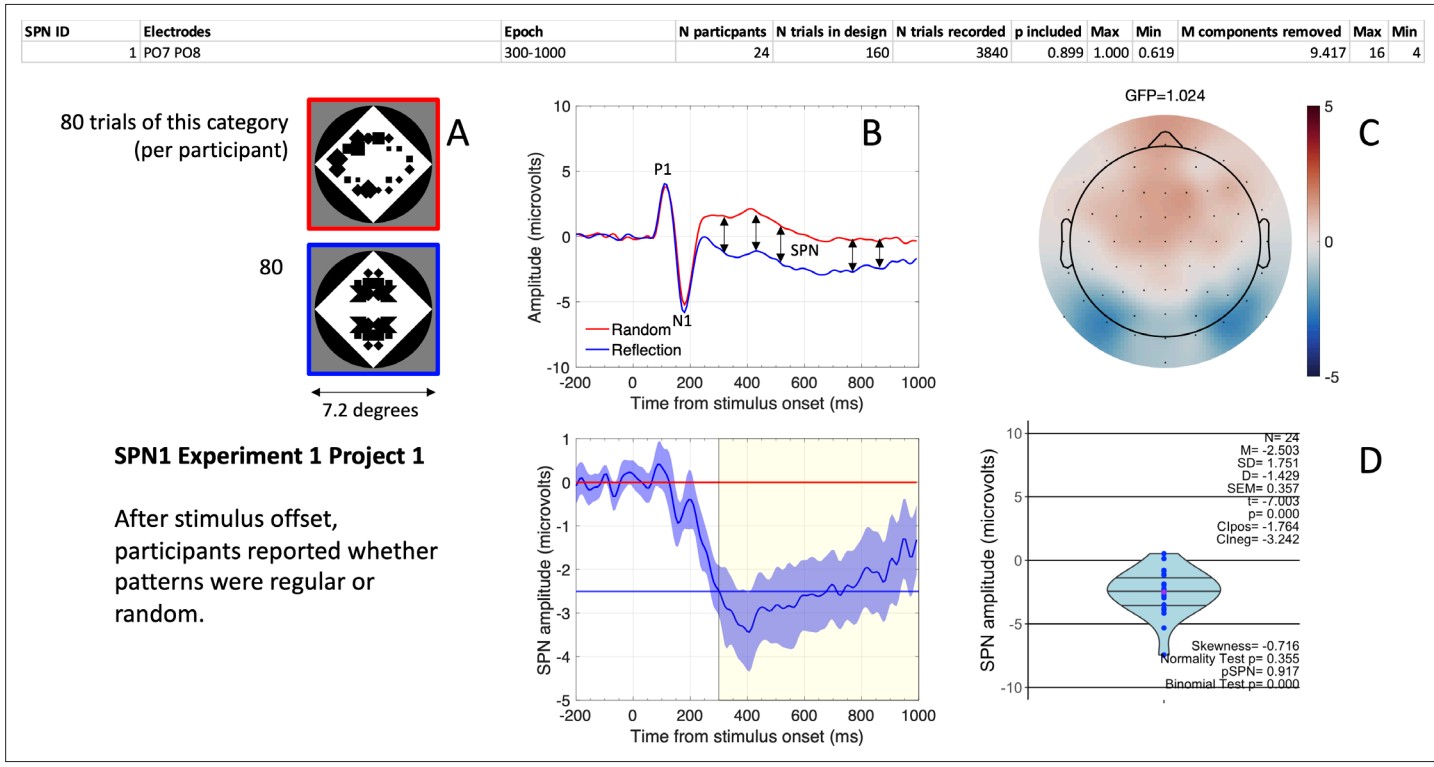

**Figure 2.** The first SPN from the SPN Gallery. (**A**) Examples of stimuli. (**B**). Grand-average ERP waves from electrodes PO7 and PO8 (upper panel), and the SPN as a reflection-random difference wave (with 95% CI; lower panel). The typical 300–1000ms SPN window is highlighted in yellow. Mean amplitude during this window was –2.503 microvolts (horizontal blue line). (**C**) SPN as a topographic difference map. (**D**) Violin plot showing SPN amplitude for each participant plus descriptive and inferential statistics. The file "One SPN Gallery.pdf" (https://osf.io/eqhd5/) contains a figure like this for all 249 SPNs. The analysis details shown at the top of the figure are also explained in this file.

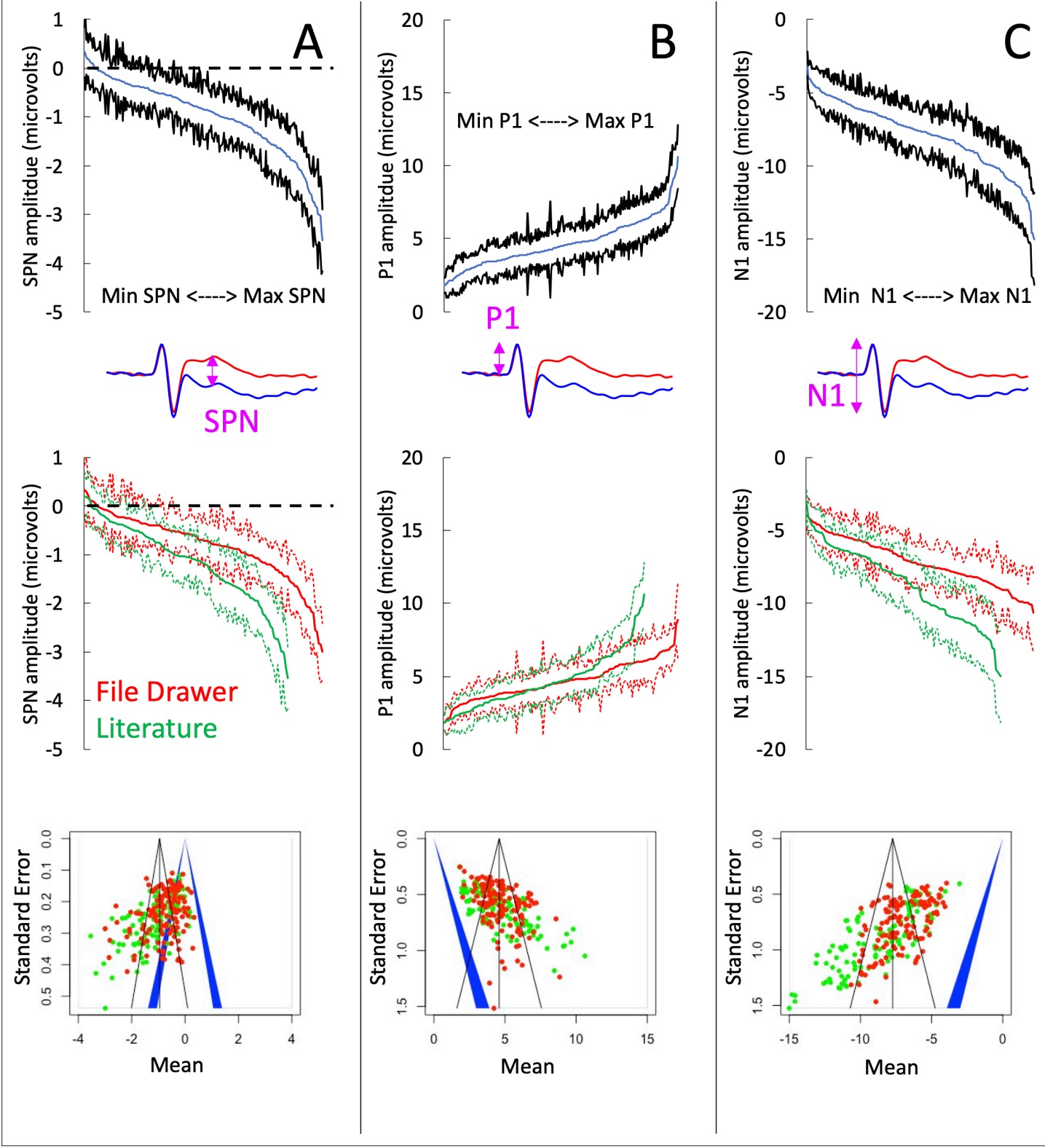

**Figure 3.** SPN amplitudes in published and unpublished work. (**A**) The top panel shows the cumulative distribution of all 249 grand-average SPNs. The smallest SPN is at the left-most end of the x-axis, and the largest SPN is at the right-most end. The blue line is comprised 249 data points, and the black lines show 95% confidence intervals. If the upper confidence interval does not rise above zero, we have a significant SPN (*P*<.05, two-tailed). The middle panel shows that the 134 unpublished SPNs in the file drawer (red) are smaller (i.e., less negative) than the 115 published SPNs in the literature (green). The bottom panel shows a funnel plot of 249 grand-average SPNs arranged by mean (x-axis) and standard error (y-axis). Red dots are unpublished SPNs,

*Figure 3 continued on next page*

*Figure 3 continued*
green dots are published SPNs. Dots to the left of the blue central triangle represent significant SPNs (inner edge, *P*<.05, outer edge, *P*<.01); if dots are inside the blue triangle, the effect is non-significant. (**B**) Equivalent set of plots for the peak amplitude P1 on regular trials. (**C**) Equivalent set of plots for the trough amplitude N1 on regular trials.

minimum effect that would still be of theoretical interest. *Brysbaert, 2019*, suggests this may often be ~0.4 in experimental psychology, and this requires 52 participants. Indeed, the theoretically interesting effect of Task on SPN amplitude could be in this range.

### Power of nonparametric tests

Of the 249 SPNs, 9.2% were not normally distributed about the mean according to the Shapiro-Wilk test (8.4% according to Kolmogorov-Smirnov test). Non-parametric statistics could thus be appropriate for some SPN analyses. For a non-parametric SPN, significantly more than half of the participants must have lower amplitude in the regular condition. We can examine this with a binomial test. Consider a typical 24 participant SPN: For a significant binomial test (*P*<.05, two tailed), we need at least 18/24=3/4 participants in the sample to show the directional effect (regular < random). Next, consider doubling sample size to 48: We now need only 32/48=2/3 participants in the sample to show the directional effect. *Figure 4D–F* illustrates the proportion of participants showing the directional effect as a function of SPN amplitude. Only 146 of the 249 grand-average SPNs (59%) were computed from a sample where at least 3/4 of the participants showed the directional effect. Meanwhile, 183 (73%) were from a sample where at least 2/3 of the participants showed the directional effect (blue horizontals in *Figure 4D*). This analysis recommends increasing sample size to 48 in future experiments.

### Power of SPN modulation effects

When there are more than two conditions, mean SPN differences may be tested with ANOVA. To assess statistical power, we reran 40 representative ANOVAs from published and unpublished work. This includes all those which support important theoretical conclusions (*Makin et al., 2015*; *Makin et al., 2016*; *Makin et al., 2020c*; see https://osf.io/hgncs/ for a full list of the experiments used in this analysis). Observed power was less than the desired 0.8 in 15 of 40 (*Figure 4H*). We note that several underpowered analyses are from Project 7 (*Makin et al., 2015*). This is still an active research area, and we will increase sample size in future experiments.

### Increasing the number of trials

Another line of attack is to increase the number of trials per participant. *Boudewyn et al., 2018* argue that adding trials is alternative way to increase statistical power in ERP research, even when split-half reliability is apparently near ceiling (as it is in SPN research: *Makin et al., 2020b*). In one highly relevant analysis, *Boudewyn et al., 2018* examined a within-participant 0.5 microvolt ERP modulation with a sample of 24 (our median sample). Increasing the number of trials from 45 to 90 increased the probability of achieving a significant effect from ~.54 to~.89 (see figure eight in *Boudewyn et al., 2018*). These authors caution that simulations of other ERP components are required to establish generalizability. We typically include at least 60 trials in each condition. However, going up to 100 trials per condition could increase SPN effect size, and this may mitigate the need to increase sample size (*Baker et al., 2021*). Of course, too many trials could introduce participant fatigue and a consequent drop in data quality. There is likely a sample size X trial number 'sweet spot' and are unlikely to have hit it already by luck.

### Typical sample sizes in other EEG research

When planning our experiments, we have often assumed that 24 is a typical sample size in EEG research. This can be checked objectively. We searched open-access EEG articles within the PubMed Central database using a text-mining algorithm. A total of 1,442 sample sizes were obtained. Mean sample size was 35 (±22.97) and a median was 28. The most commonly occurring sample size was 20. We also extracted sample sizes from 74 EEG datasets on the OpenNeuro BIDS compliant repository. The mean sample size was 39.34 (±38.56), the median was 24, and the mode was again 20. Our SPN experiments do indeed have typical sample sizes, as we had assumed.

### Summary for horseman two: low statistical power

Low statistical power is an obstacle to cumulative SPN research. Before the COVID pandemic stopped all EEG research, we were completing

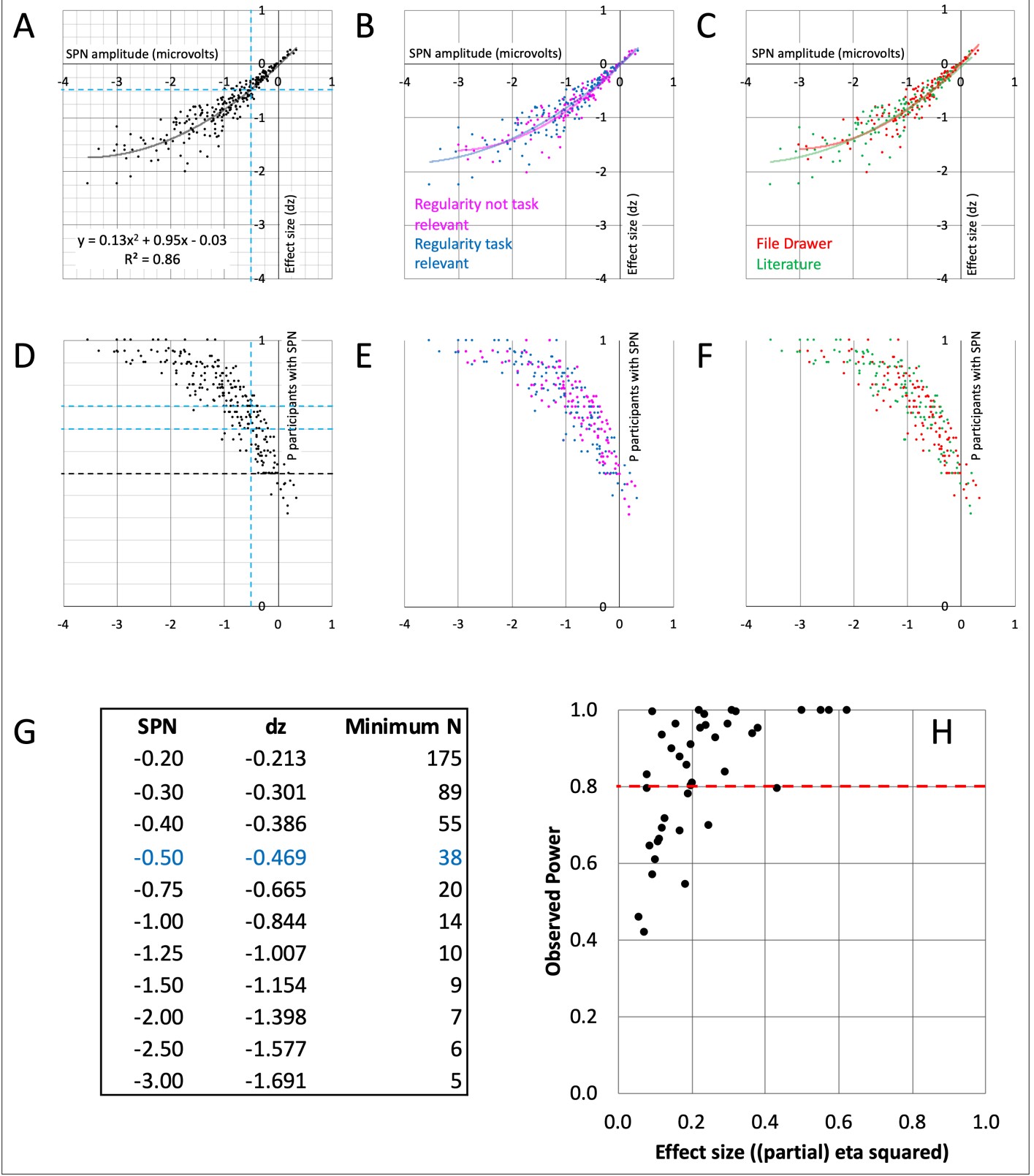

**Figure 4.** SPN effect size and power. (**A–C**) The nonlinear relationship between SPN amplitude and effect size. The equation for the second order polynomial trendline is shown in panel A ($y = 0.13x^2 + 0.95 x - 0.03$). This explains 86% of variance in effect size ($R^2=0.86$). Using the equation, we can estimate effect size for an SPN of a given amplitude. Dashed lines highlight –0.5 microvolt SPNs, with average effect size $d_z$ of –0.469. For an 80% chance of finding this effect, an experiment requires 38 participants. The relationships were similar whether regularity was task relevant or not (**B**), and in

*Figure 4 continued on next page*

*Figure 4 continued*

published and unpublished work (**C**). (**D–F**) How many participants show the SPN? The larger (more negative) the SPN, the more individual participants show the effect (regular < random). Dashed lines highlight –0.5 microvolt SPNs, which are quite often present in 2/3 but not 3/4 of the participants. The relationships were similar whether regularity was task relevant or not (**E**), and in published and unpublished work (**F**). (**G**) Table of required N for 80% chance of obtaining an SPN of a given amplitude. (**H**) Observed power and effect size of 40 SPN modulations. 15/40 do not reach the 0.8 threshold (red line).

around 10 EEG experiments per year with a median sample of 24. When EEG research starts again, we may reprioritize, and complete fewer experiments per year with more participants in each. Furthermore, one can tentatively assume that other ERPs have comparable signal/noise properties to the SPN. If so, we can plausibly infer that many ERP experiments are under-powered for detecting 0.5 microvolt effects. *Figure 4G* thus provides a rough sample size guide for ERP researchers, although we stress that more ERP-specific estimates of effect size should always be treated as superior. We also stress that our sample size recommendations do not directly apply to multi-level or multivariate analyses, which are increasingly common in many research fields. Nevertheless, this investigation has strengthened our conviction that EEG researchers should collect larger samples by default. Several benefits more than make up for the extra time spent on data collection. Amongst other things, larger samples reduce Type 2 error, give more accurate estimates of effect size, and facilitate additional exploratory analyses on the same data sets.

## Horseman three: P-hacking

Sensitivity of an effect to arbitrary analytical options is called the 'vibration of the effect' (*Button et al., 2013*). An effect that vibrates substantially is vulnerable to 'P-hacking': that is, exploiting flexibility in the analysis pipeline to nudge effects over the threshold for statistical significance (for example, *P*=.06 conveniently becomes *P*=.04, and the result is publishable). "Double dipping" is one particularly tempting type of P-hacking for cognitive neuroscientists because we typically have such large, multi-dimensional data sets (*Kriegeskorte et al., 2009*). Researchers can dip into a large dataset, observe where something is happening, then run statistical analysis on this data selection alone. For example, in one early SPN paper, *Höfel and Jacobsen, 2007a* state that "Time windows were chosen after inspection of difference waves" (page 25, section 2.8.3). It is commendable that Höfel and Jacobsen were so explicit: often

double dipping is spread across months where researchers alternate between 'preliminary' data visualization and 'preliminary' statistical analysis so many times that they lose track of which came first. Double dipping beautifies results sections, but without appropriate correction, it inflates Type 1 error rate. We thus attempted to estimate the extent of P-hacking in our SPN research, with a special focus on double dipping.

### Electrode choice

Post hoc electrode choice can sometimes be justified: Why analyse an electrode cluster that misses the ERP of interest? Post hoc electrode choice could be classed as a *questionable research practice* rather than flagrant malprac-tice (*Agnoli et al., 2017*; *Fiedler and Schwarz, 2015*; *John et al., 2012*). Nevertheless, we must at least assess the consequences of this flexibility. What would have happened if we had dogmat-ically stuck with the same electrodes for every analysis, without granting ourselves any flexibility at all?

To estimate this, we chose three a priori bilat-eral posterior electrode clusters and recomputed all 249 SPNs (Cluster 1 = [PO7 O1, O2 PO8], Cluster 2 = [PO7, PO8], Cluster 3 = [P1 P3 P5 P7 P9 PO7 PO3 O1, P2 P4 P6 P8 P10 PO8 PO4 O2]). The first two clusters were chosen because we have used them often in published research (Cluster 1: *Makin et al., 2020c*; Cluster 2: *Makin et al., 2016*). Cluster 3 was chosen because the 16 electrodes cover the whole bilateral posterior region. These three clusters are labelled on a typical SPN topoplot in *Figure 5A*. Reassuringly, we found that SPN amplitude is highly correlated across clusters (Pearson's *r* ranged from .946 to .982, *P*<.001; *Figure 5B*). This suggests vibra-tion is low, and flexible electrode choice has not greatly influenced our SPN findings. In fact, mean SPN amplitude would have been slightly higher if we had used Cluster 2 for all projects. Average SPN amplitude was –0.98 microvolts [95% CI = –1.08 to –0.88] with the original cluster, –0.961 [–1.05; –0.87] microvolts for Cluster 1,–1.12 [–1.22; –1.01] microvolts for Cluster 2, and –0.61 [–0.66; –0.55] microvolts for Cluster 3.

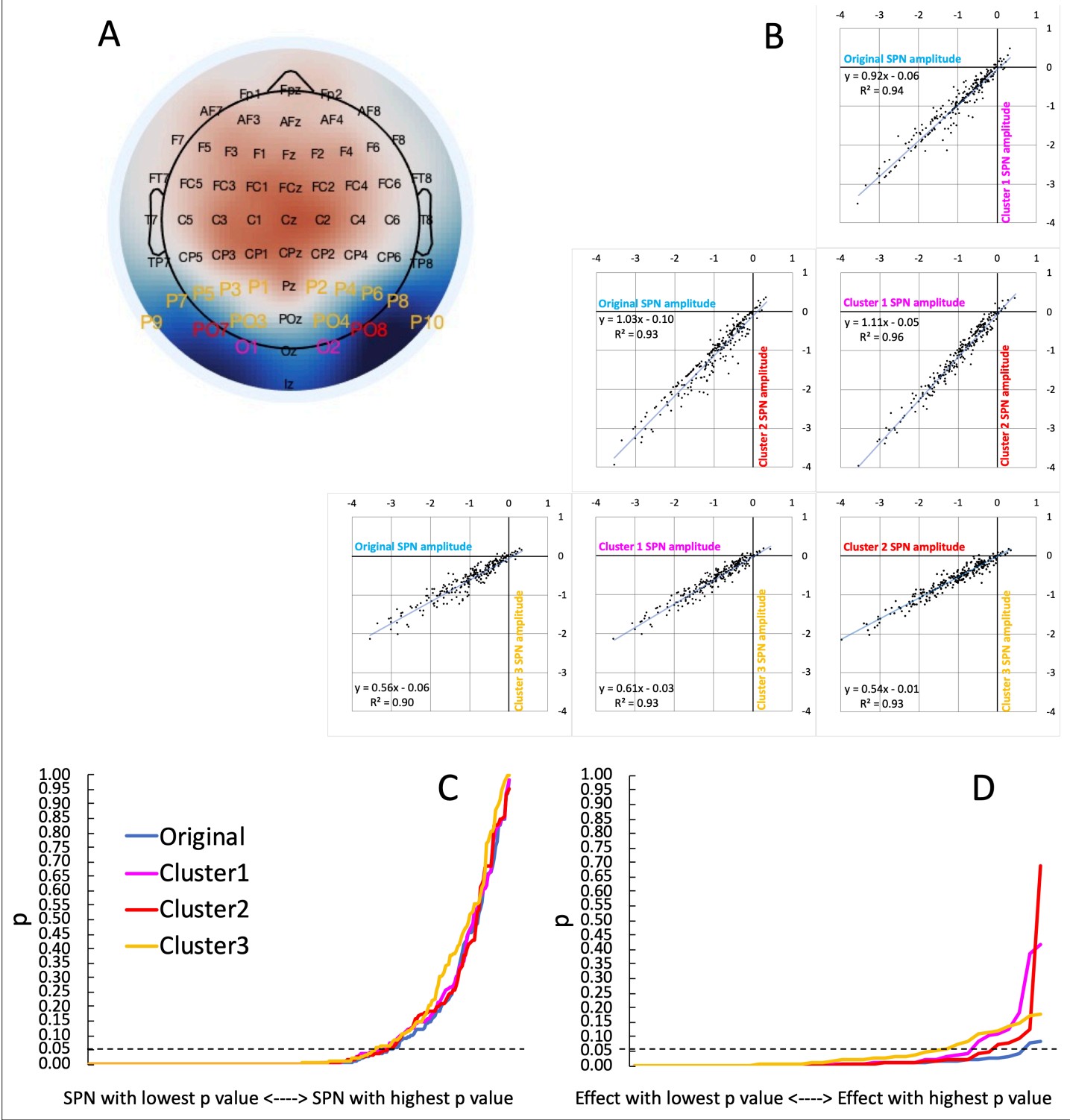

**Figure 5.** Vibration of the SPN effect. (**A**) Typical SPN topographic difference map with labels colour coded to show three alternative electrode clusters, which could have been used dogmatically in all analyses, whatever the observed topography. (**B**) Scatterplots show SPNs from the original cluster and the three alternatives, which are highly correlated. (**C**) One-sample t-tests were used to establish whether each SPN is significant. The cumulative distribution of p values is shown here. The smallest p value (from the most significant SPN) is at the left-most end of the x axis, and the largest p value (from the least significant SPN) is at the right-most end. The $p$ values from the original cluster and the three alternatives were very similar. There was a similar number of significant SPNs (169–177). (**D**) ANOVAs are used to assess SPN modulations. The $p$ values from 40 representative ANOVA effects do not overlap completely. There were more significant SPN modulations when the original electrode cluster was used than any alternative (38 vs 35–29).

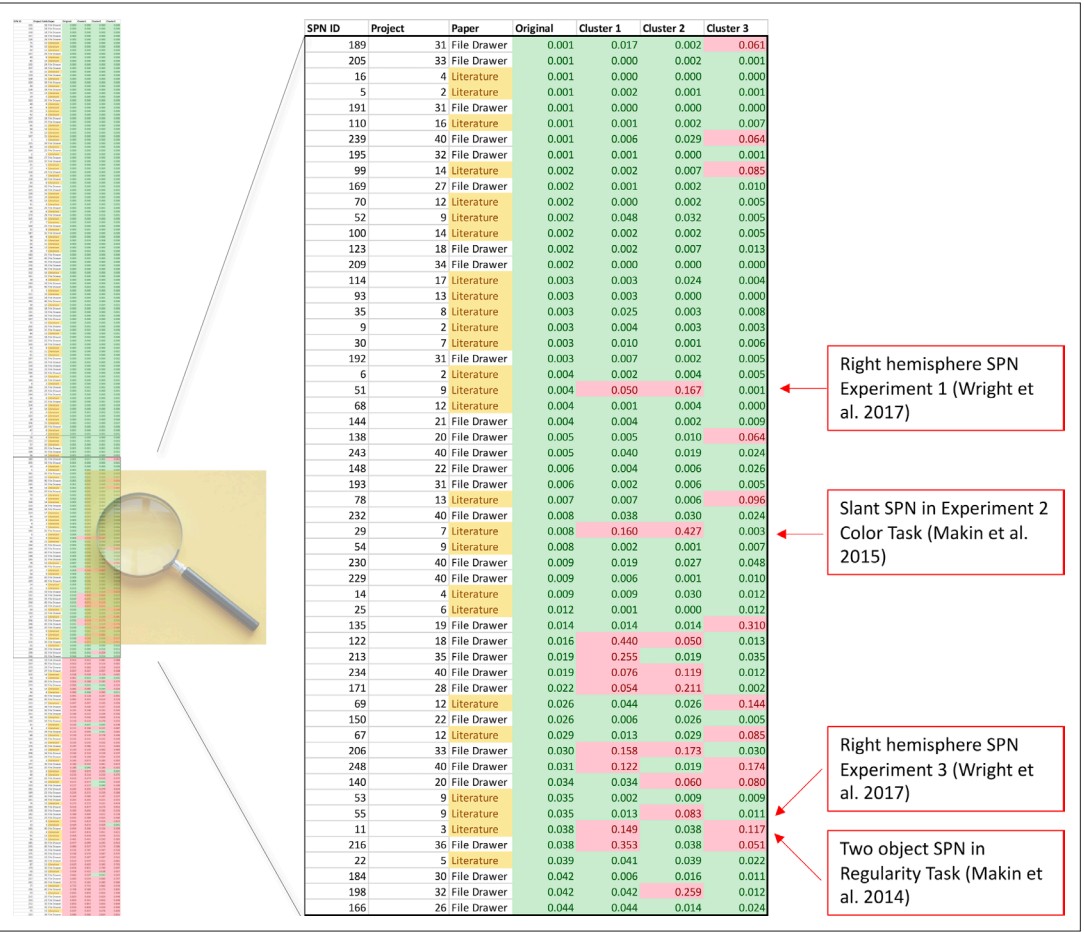

**Figure 6.** Which SPNs are significant using alternative clusters? The left column shows a table of all 249 SPNs, colour coded (green, significant; red, non-significant), and sorted by *p* value obtained using the original cluster. The important part of the table is the centre, where significance thresholds are crossed by some clusters but not others. The central part is expanded, so text is now readable. The important cases are published SPNs that are not significant when either Cluster 1 or 2 is used instead. These 4 cases are all labelled (red boxes).

Next, we ran one-sample t-tests on the SPN as measured at the 4 alternative clusters. The resulting p values are shown cumulatively in *Figure 5C*. Crucially, the area under the curve is similar for all cases. The significant SPN count varies only slightly, between 169 and 177 ($\chi^2$ (3)=0.779, *P*=.854). We conclude that flexible electrode choice has not substantially inflated the number of significant SPNs in our research.

To illustrate this in another way, *Figure 6* shows a colour-coded table of *p* values, sorted by original cluster. At the top there are many rows which are significant whichever cluster is used (green rows). At the bottom there are many rows which are non-significant whichever cluster is used (red rows). The interesting rows are in the middle, where there is some disagreement indicating that the original effect was a false positive (some green and some red cells on each row). We can zoom in on the central portion: where,

exactly, are the disagreements? Two cases come from *Wright et al., 2017* however, this project reported a contralateral SPN, and these are inevitably more sensitive to electrode choice because they only cover half the scalp surface area.

We applied the same reanalysis to our 40 representative ANOVA main effects and interactions. Here there is more cause for concern: 38 of the 40 effects were significant using the original electrode cluster, however this goes down to 33 with Cluster 1, 35 with Cluster 2, and to just 29 with Cluster 3 (*Figure 5D*). Flexible electrode choice has thus significantly increased the number of significant SPN modulations ($\chi^2$ (3)=8.107, *P*=.044).

## Spatio-temporal clustering
The above analysis examines consequences of choosing different electrode clusters a priori, while holding time window constant. Next, we

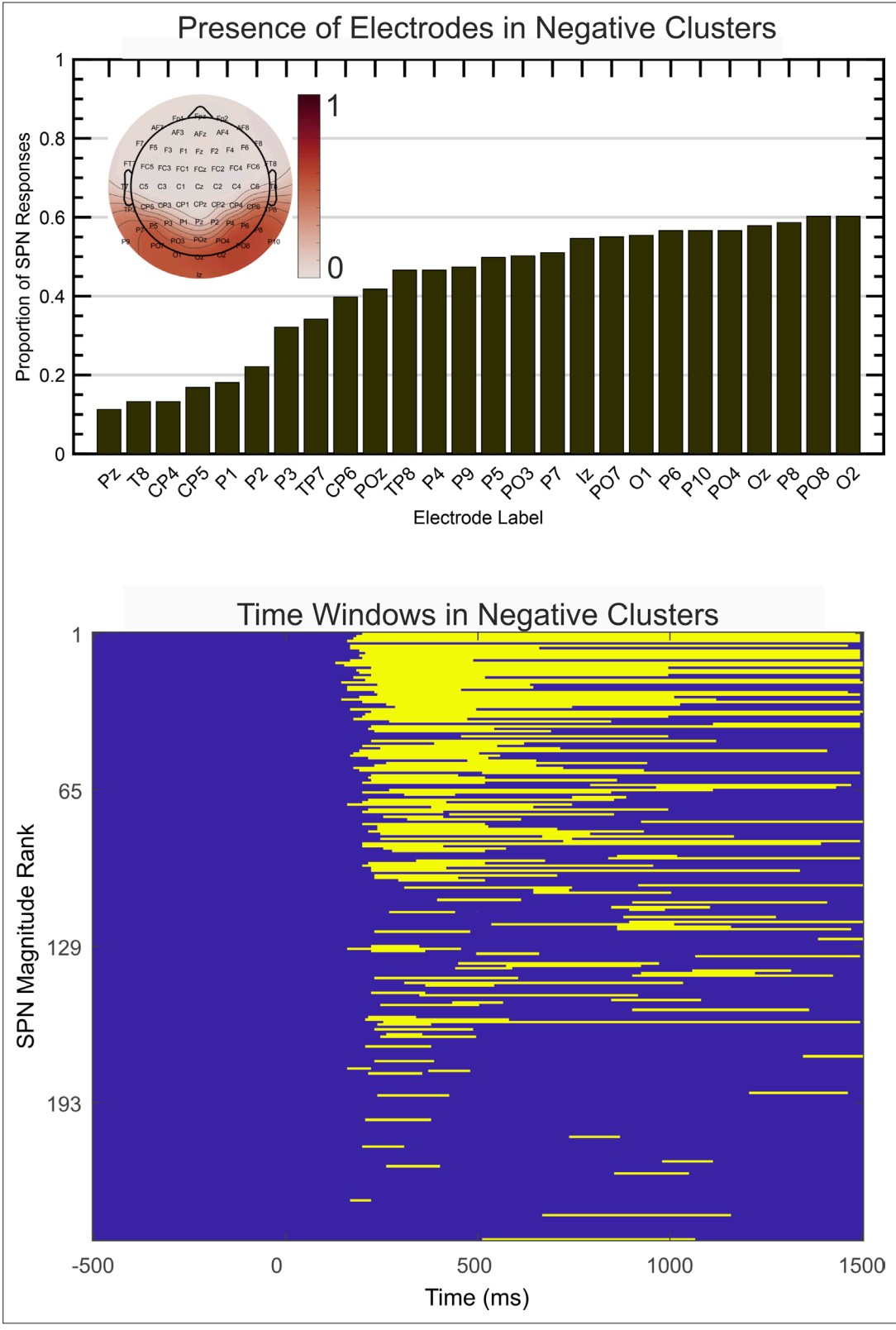

**Figure 7.** Spatio-temporal clustering results. The upper image illustrates the proportion of times each electrode appeared in the most significant negative cluster. Electrodes appearing in less than 10% of cases are excluded. The topoplot inset shows proportions on a colour scale for all electrodes. The lower image illustrates the time course over which the same negative clusters were active, ranked by SPN magnitude.

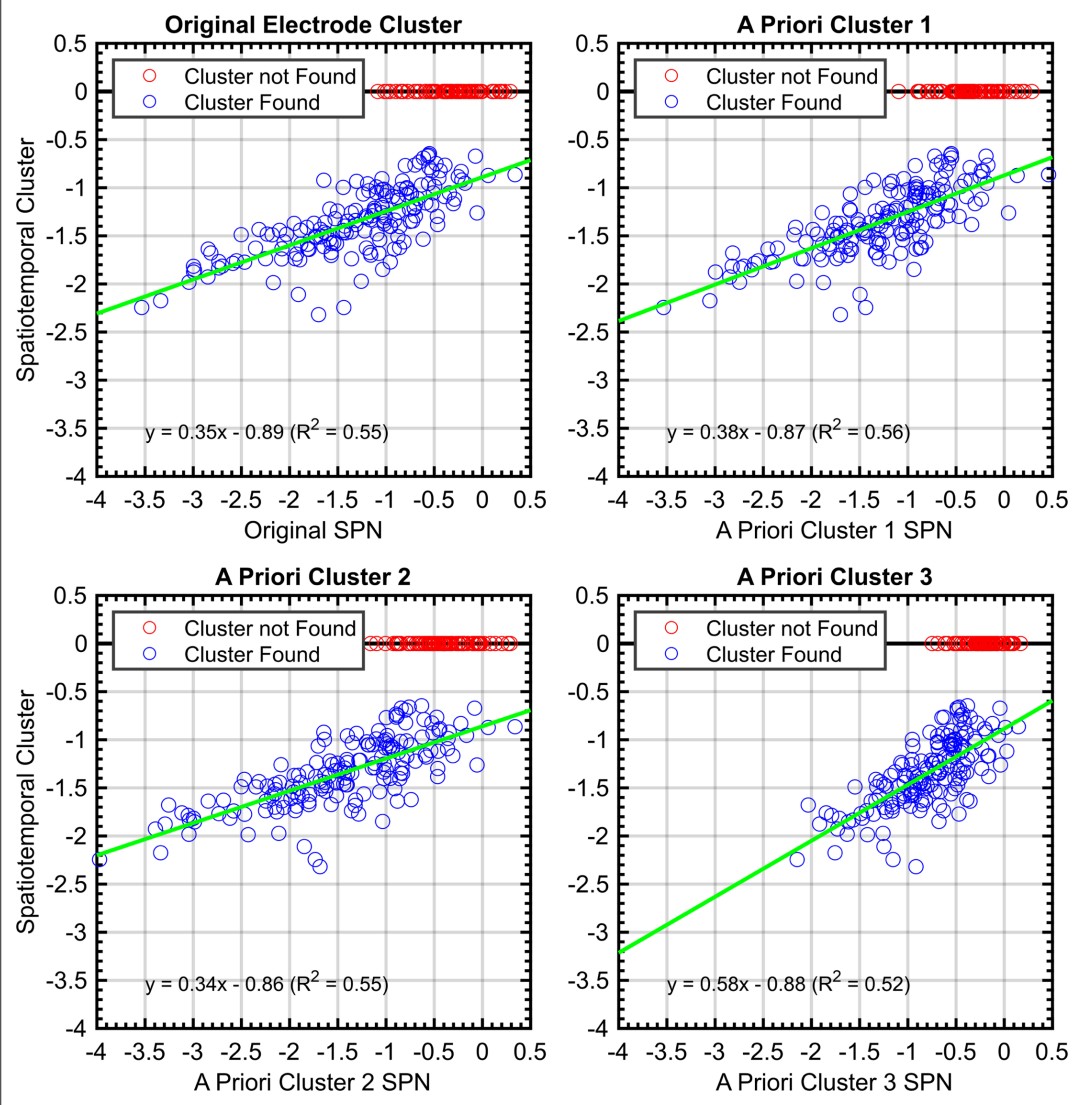

**Figure 8.** Correlations between SPNs from a priori and data-driven selections. Red data points indicate no significant negative cluster was found for that SPN. For these points, the mean SPN cluster is plotted as zero and does not influence the green least-squares regression line.

used a spatio-temporal clustering technique (*Maris and Oostenveld, 2007*) to identify both electrodes and timepoints where the difference between regular and irregular conditions is maximal. Does this purely data driven approach lead to the same conclusions as the original analysis from a priori electrode clusters?

After obtaining all negative electrode clusters with significant effect ($P<.05$, two tailed) the single most significant cluster was extracted for each SPN. The proportion of times that each electrode appeared in this cluster is illustrated in *Figure 7A* (electrodes present in less than 10% of cases are excluded). Findings indicate that electrodes O1 and PO7 are most likely to capture the SPN over the left hemisphere, and O2 and PO8

are most likely to capture the SPN over the right hemisphere. This is consistent with our typical a priori electrode selections. *Figure 7B* shows activity was mostly confined to the 200 to 1000ms window, extending somewhat to 1500ms. This is consistent with our typical a priori time window selections. Apparently there were no effects at electrodes or at time windows we had previously neglected.

To quantify these consistencies, SPNs were recomputed using the electrode cluster and time window obtained with spatiotemporal cluster analysis. There was a strong correlation between this and SPN from each a priori cluster (Pearson's $r$ ranged from .719 to .746, $P<.001$; *Figure 8*).

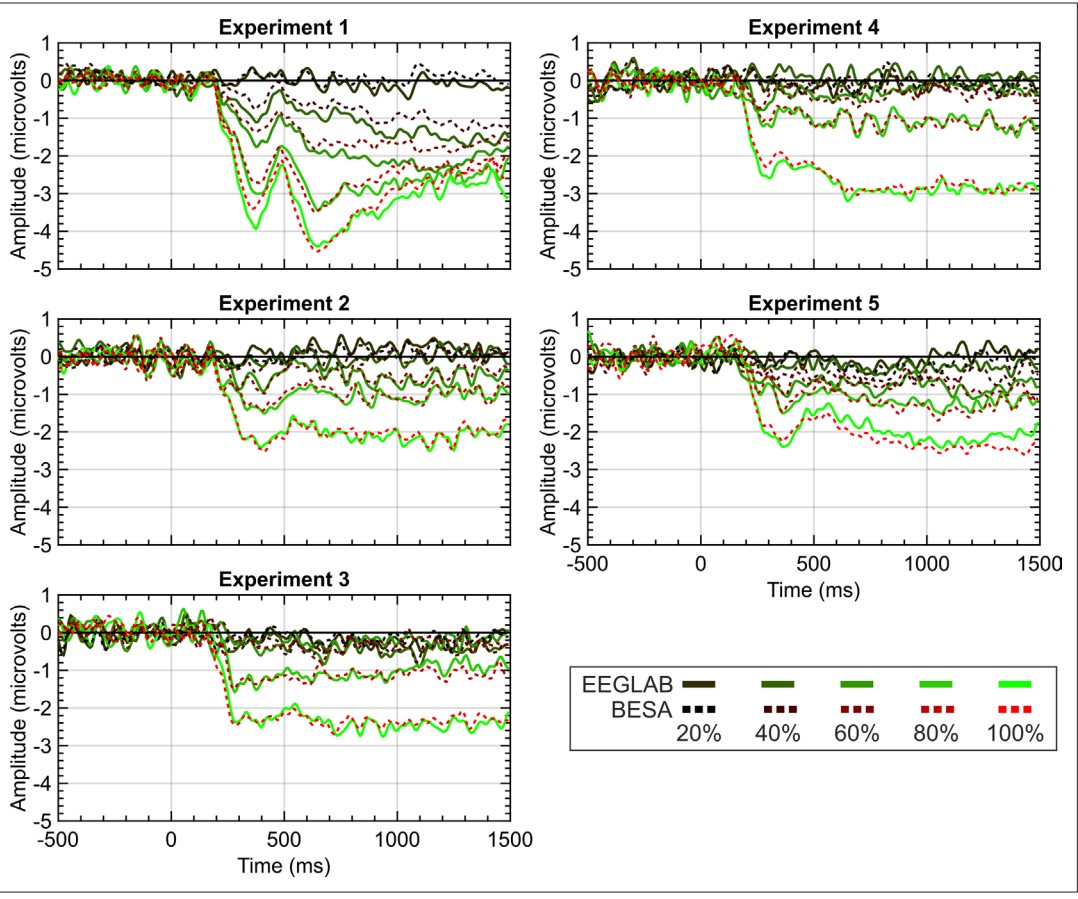

**Figure 9.** EEGLAB and BESA pipeline comparison. Panels show SPN waves for each of the 5 experiments in project 13. EEGLAB waves are the solid lines; the BESA waves are the dashed lines. 20%–100% refers to the proportion of symmetry in the stimulus (see *Figure 1* for example stimuli).

There are two further noteworthy results from spatiotemporal clustering analysis. First, 74% of published data sets yielded a significant cluster, while the figure was only 57% for unpublished data (as of May 2022). This is another estimate of publication bias. Second, we can explain some of the amplitude variation shown in *Figure 8*. As described in *Box 2*, 33% of variance in grand-average SPN amplitude can be predicted by two factors called W and Task (SPN (microvolts) = –1.669 W – 0.416Task + 0.071). We also ran this regression analysis on the spatiotemporal cluster SPNs (i.e., the blue data points in *Figure 8*). The two predictors now explained just 16.8% of variance in grand-average SPN amplitude (SPN (microvolts) = –0.557 W – 0.170Task – 0.939). The reduced $R^2$, shallower slopes and lower intercept are largely caused by the fact that many data sets had to be excluded because they did not yield a cluster (i.e., the red data points in *Figure 8*). This highlights one disadvantage of spatiotemporal clustering: For many purposes we want to include small and non-significant SPNs in pooled analysis of the whole catalogue. However, these relevant data points are missing or artificially set at zero if SPNs are extracted by spatiotemporal clustering.

### Pre-processing pipelines

There are many pre-processing stages in EEG analysis (*Pernet et al., 2020*). For example, our data is often re-referenced to a scalp average, low pass filtered at 25 Hz, down-sampled to 128 Hz and baseline corrected using a –200 to 0ms pre-stimulus interval. We then remove some blink and other large artifacts with independent components analysis (*Jung et al., 2000*). We sometimes remove noisy electrodes with spherical interpolation. We then exclude trials where amplitude exceeds +/-100 microvolts at any electrode.

To examine whether these pre-processing norms are consequential, we reanalysed data from 5 experiments from Project 13 in BESA instead of Matlab and EEGLAB, using different cleaning and artifact removal conventions. When using BESA, we employed the recommended

pipelines and parameters. Specifically, we used a template matching algorithm to identify eye-blinks and used spatial filtering to correct eye-blinks within the continuous data. Trials were removed that exceeded an amplitude of ±120 microvolts or a gradient of ±75 (with the gradient being defined as the difference in amplitude between two neighbouring samples). Although this trial exclusion takes place on filtered data, remaining trials are averaged across pre-filtered data. High and low pass filters were set to 0.1 and 25 Hz in both EEGLAB and BESA. While EEGLAB used zero-phase filters, filtering in BESA used a forward-filter for the high-pass filter but used a zero-phase filter for the low-pass. As seen in *Figure 9*, similar grand-average SPNs fall out at the end of these disparate pipelines.

We conclude that post-hoc selection of electrodes and time windows are weak points that can be exploited by unscrupulous P-hackers. Earlier points in the pipeline are less susceptible to P-hacking because changing them does not predictably increase or decrease the desired ERP effect.

***Summary for horseman three: P-hacking***
It is easier to publish a simple story with a beautiful procession of significant effects. This stark reality may have swayed our practice in subtle ways. However, our assessment is that P-hacking is not a pervasive problem in SPN research, although some effects rely too heavily on post hoc data selection.

Reassuringly, we found most effects could be recreated with a variety of justifiable analysis pipelines: Data-driven and a priori approaches gave similar results, as did different EEGLAB and BESA pipelines. This was not a foregone conclusion – related analysis in fMRI has revealed troubling inconsistencies (*Lindquist, 2020*; *Botvinik-Nezer et al., 2020*). It is advisable that all researchers compare multiple analysis pipelines to assess vibration of effects and calibrate their confidence accordingly.

## Horseman four: HARKing
Hypothesizing After Results Known or HARKing (*Kerr, 1998*; *Rubin, 2017*), is a potential problem in EEG research. Specifically, it is possible to conduct an exploratory analysis, find something unexpected, then describe the results *as if* they were predicted a priori. At worst, a combination of P-hacking and HARKing can turn noise into theory, which is then cited in the literature for decades. Even without overt HARKing, one can

*beautify* an introduction section after the results are known, so that papers present a simple narrative (maybe this post hoc beautification could be called BARKing). Unlike the other horses, HARKing and BARKing are difficult to diagnose with reanalysis, which is why this section is shorter than the previous sections.

The main tool to fight HARKing is online pre-registration of hypotheses (for example, on aspredicted.org or osf.io). We have started using pre-registration routinely in the last four years, but we could have done so earlier. An even stricter approach is to use registered reports, where the introductions and methods are peer reviewed before data collection. This can abolish both HARKing and BARKing (*Chambers, 2013*; *Munafò et al., 2017*), but we have only just started with this. Our recommendation is to use heavy pre-registration to combat HARKing. Perhaps unregistered EEG experiments will be considered unpublishable in the not-too-distant future.

## Discussion
One of the most worrisome aspects of the replication crisis is that problems might be systemic, and not caused by a few corrupt individuals. It seems that the average researcher publishes somewhat biased research, without sufficient self-awareness. So, what did we find when we looked at our own research?

As regards publication bias – the first horseman of irreproducibility – we found that the 115 published SPNs were slightly stronger than the 134 unpublished ones. However, we are confident that there is no strong file drawer problem here. Even the unpublished SPNs are in the right direction (regular < random not random < regular). Furthermore, a complete SPN catalogue itself fights the consequences of publication bias by placing everything that was in the file drawer into the public domain.

We are more troubled by the second horseman: low statistical power. Our most negative conclusion is that reliable SPN research programs require larger samples than those we typically obtain (38 participants are required to reliably measure –0.5 microvolt SPNs and our median sample size is 24). This analysis has lessons for all researchers: It is evidently possible to 'get by' while routinely conducting underpowered experiments. One never notices a glaring problem: after all, underpowered research will often yield a lucky experiment with significant results, and this may support a new publication

before moving on to the next topic. However, this common practice is not a strong foundation for cumulative research.

The costs of underpowered research might be masked by the third horseman: P-hacking. Researchers often exploit flexibility in the analysis pipeline to make borderline effects appear significant (*Simmons et al., 2011*). In EEG research, this often involves post-hoc selection of electrodes and time windows. Although some post-hoc adjustment is arguably appropriate, this double dipping certainly inflates false positive rate and requires scrutiny. We found that the same basic story would have emerged if we had rigidly used the same a priori electrode clusters in all projects or used a spatio-temporal clustering algorithm for selection. However, some of our SPN modulations were not so robust, and we have relied on post hoc time windows.

The fourth horseman, HARKing, is the most difficult dimension to evaluate because it cannot be diagnosed with reanalysis. Nevertheless, pre-registration is the best anti-HARKing tool, and we could have engaged with this earlier. We are just beginning with pre-registered reports.

To summarize these evaluations, we would tentatively self-award a grade of A- for publication bias (75%, or lower first class, in the UK system), C+ for statistical power (58%), B+ for P-Hacking (68%), and B for Harking (65%). While some readers may not be interested in the validity of SPN research per se, they may be interested in this meta-scientific exercise, and we would encourage other groups to perform similar exercises on their own data. In fact, such exercises may be essential for cumulative science. It has been argued that research should be auditable (*Nelson et al., 2018*), but Researcher A will rarely be motivated to audit a repository uploaded by Researcher B, even if the datasets are FAIR. To fight the replication crisis, we must actively look in the mirror, not just passively let others snoop through the window.

*Klapwijk et al., 2021* have performed a similar meta-scientific evaluation in the field of developmental neuroimaging and made many practical recommendations. All our themes are evident in their article. We also draw attention to international efforts to estimate the replicability of influential EEG experiments (*Pavlov et al., 2020*). These mass replication projects provide a broad overview. Here we provide depth, by examining all the data and practices from one representative EEG lab. We see these approaches as complementary: they both provide insight into whether a field is working in a way that generates meaningful results.

The focus on a single lab inevitably introduces some biases and limitations. Other labs may use different EEG apparatus with more channels. This would inevitably have some effect on the recordings. More subtle differences may also matter: For instance, other labs may use more practice trials or put more stress on the importance of blink suppression. However, the heterogeneity of studies and pipelines explored here ensures reasonable generalizability. We are confident that our conclusions are relevant for SPN researchers with different apparatus and conventions.

Curated databases are an extremely valuable resource, even if they are not used for meta-scientific evaluation. Public catalogues are an example of large-scale neuroscience, the benefits of which have been summarized by the neuroscientist Jeremy Freeman as follows: "Understanding the brain has always been a shared endeavour. But thus far, most efforts have remained individuated: labs pursuing independent research goals, slowly disseminating information via journal publications, and when analyzing their data, repeatedly reinventing the wheel" (*Freeman, 2015*). We tried to make some headway here with the SPN catalogue.

Perhaps future researchers will see their role as akin to expert museum curators, who oversee and update their public catalogues. They will obsessively tidy and perfect the analysis scripts, databases and metafiles. They will add new project folders every year, and judiciously determine when previous theories are no longer tenable. Of course, many researchers already dump raw data in online repositories, but this is not so useful. Instead, we need FAIR archives which are actively maintained, organized, and promoted by curators. The development of software, tools and shared repositories within the open science movement is making this feasible for most labs. We are grateful to everyone who is contributing to this enterprise.

It took more than a year to find all the SPN data, organize it, reformat it, produce uniform scripts, conduct rigorous double checks, and upload material to public databases. However, we anticipate that it will save far more than a year in terms of improved research efficiency. It is also satisfying that unpublished data sets are not merely lost. Instead, they are now contributing to more reliable estimates of SPN effect size and power. It is unlikely that any alternative activity could have been more beneficial for SPN research.

## Acknowledgements

Many people contributed to the complete Liverpool SPN catalogue. One notable past contributor was Dr Letizia Palumbo (who was a postdoc 2013–2016). Many undergraduate, postgraduate and intern students have also been involved in data collection.

**Alexis DJ Makin** is in the Department of Psychological Sciences, University of Liverpool, Liverpool, United Kingdom

alexis.makin@liverpool.ac.uk

0000-0002-4490-7400

**John Tyson-Carr** is in the Department of Psychological Sciences, University of Liverpool, Liverpool, United Kingdom

0000-0003-3364-2184

**Giulia Rampone** is in the Department of Psychological Sciences, University of Liverpool, Liverpool, United Kingdom

0000-0002-2710-688X

**Yiovanna Derpsch** is in the Department of Psychological Sciences, University of Liverpool, Liverpool, and the School of Psychology, University of East Anglia, Norwich, United Kingdom

**Damien Wright** is in the Patrick Wild Centre, Division of Psychiatry, Royal Edinburgh Hospital, University of Edinburgh, Edinburgh, United Kingdom

0000-0002-9105-3559

**Marco Bertamini** is in the Department of Psychological Sciences, University of Liverpool, Liverpool, United Kingdom, and the Dipartimento di Psicologia Generale, Università di Padova, Padova, Italy

0000-0001-8617-6864

*Author contributions:* Alexis DJ Makin, Conceptualization, Data curation, Formal analysis, Funding acquisition, Investigation, Methodology, Project administration, Resources, Software, Supervision, Validation, Visualization, Writing – original draft, Writing – review and editing; John Tyson-Carr, Data curation, Formal analysis, Investigation, Methodology, Project administration, Resources, Software, Validation, Visualization, Writing – review and editing; Giulia Rampone, Investigation, Project administration, Resources, Writing – review and editing; Yiovanna Derpsch, Investigation, Resources, Writing – review and editing; Damien Wright, Investigation, Resources, Writing – review and editing; Marco Bertamini, Conceptualization, Investigation, Project administration, Resources, Supervision, Writing – original draft, Writing – review and editing

*Competing interests:* The authors declare that no competing interests exist.

*Ethics:* This was meta-analysis of previously published work (and unpublished work). All datasets were electrophysiological recordings taken from human participants. All experiments had local ethics committee approval and were conducted according to the Declaration of Helsinki (Revised 2008). Therefore all participants gave informed consent.

## Funding

| Funder | Grant reference number | Author |
|---|---|---|
| Economic and Social Research Council | ES/S014691/1 | Alexis DJ Makin John Tyson-Carr Giulia Rampone Marco Bertamini |

The funders had no role in study design, data collection and interpretation, or the decision to submit the work for publication.

## Decision letter and Author response

Decision letter https://doi.org/10.7554/eLife.66388.sa1
Author response https://doi.org/10.7554/eLife.66388.sa2

# Additional files

## Supplementary files

• Transparent reporting form

## Data availability

All data that supports analysis and Figures, along with codes for analysis, are available on open science framework (https://osf.io/2sncj/). To make it possible for anybody to analyze our data, we developed an app that allows users to: (i) view the data and summary statistics as they were originally published; (ii) select data subsets, electrode clusters, and time windows; (iii) visualize the patterns; (iv) export data for further statistical analysis. This repository and app will be expanded to accommodate data from future projects. The app is available to download for Windows users at https://github.com/JohnTyCa/The-SPN-Catalogue (copy archived at swh:1:rev:75e729f867c275433b68807bc3f-2228c57a3ccac).

The following dataset was generated:

| Author(s) | Year | Dataset URL | Database and Identifier |
|---|---|---|---|
| Makin A | 2021 | https://osf.io/2sncj/ | Open Science Framework, 2sncj |

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
