## [Decision Letter]

**Decision letter after peer review:**

Thank you for submitting your article "Meta-Research: Lessons from a catalogue of 6674 brain recordings" to *eLife* for consideration as a Feature Article. Your article has been reviewed by three peer reviewers, and the evaluation has been overseen by the *eLife* Features Editor (Peter Rodgers). The following individuals involved in review of your submission have agreed to reveal their identity: Peter Kohler (Reviewer #1); Cyril Pernet (Reviewer #3).

The reviewers and editor have discussed the reviews and drafted this decision letter to help you prepare a revised submission. While the reviewers were largely positive about your manuscript, there are quite a few important points that need to be addressed to make the article suitable for publication.

Summary

This manuscript comes from a group in Liverpool who for the last decade have used event-related electroencephalography (EEG) to measure a brain response known as the sustained posterior negativity (SPN). The SPN provides a readout of processing of symmetry and other types of visual regularity. The goal of the current paper is not to offer new information about the role of symmetry in visual perception, but rather to provide a meta-analysis of scientific practice based on the thousands of studies, published and unpublished, accumulated by the authors over the years. The authors use this rare dataset to evaluate their own performance on four key concerns that are known to lead to irreproducible results. The conclusions the authors make about their own scientific practice are largely supported by the data, and the findings offer some recommendations on how to best measure the SPN in future studies. The authors succeed in providing an even-handed estimate of the reproducibility of their research over the last 10 years but are less successful in generalizing their conclusions.

The main strength of the manuscript is the data set, which contains every SPN study ever recorded at the University of Liverpool – more than 6000 recordings of the SPN from more than 2000 participants. The authors have obviously put substantial effort into cleaning up and organizing the data and have made the dataset publicly available along with analysis tools. As such, the data represents a herculean exercise in data sharing and open science and should inspire EEG researchers who are interested in following best practices in those domains. Another strength of the paper are the analyses of publication bias and statistical power, which are quite well-done and fairly unique, at least for EEG research.

The dataset clearly has value in itself, but the potential broader interest of the manuscript depends critically on what the authors do with the data and to what extent their findings can provide general recommendations. The main weakness is that the findings are fairly specific to the measurement modality and brain response (SPN) measured by the authors. In my reading there is generally little attempt to relate the conclusions to other event-related EEG signals, much less to other EEG approaches or other measurement modalities. The sections on Publication Bias and Low Statistical Power are arguably inherently difficult to generalize beyond the SPN. The section on P-hacking, however, is directly limited in impact by analyses that are either very simplistic or missing entirely. The electrode choice section attempts to evaluate the impact of that particular approach to dimensionality reduction, but that is only one of many approaches regularly used in EEG research. The authors discuss that the selection of time window is an important parameter that can influence the results, but there is no actual analysis of the effect of time window selection across the many studies in the dataset. Finally, the analysis of the effect of pre-processing in the same section is also fairly limited in impact, because only one experiment is reanalysed, rather than the whole dataset.

In conclusion, the manuscript will clearly be of interest for the relatively narrow range of researchers who study the SPN using event-related EEG. In addition, the dataset will serve as a useful example of large-scale curated sharing of EEG data. Beyond that, I think the impact and utility of the manuscript in its current from will be relatively limited.

Essential Revisions

1 Data sharing

We encourage the authors to make their data available in a way that is FAIR. As it stands the data are Findable and Accessible, but not much Interoperable or Reusable. For example, using BIDS elecrophysiological derivatives and given the authors experience with EEGLAB, they could use the.set format along with a flat structure and proper metadata (particiants.tsv, data_description.json, channels.tsv, electrodes.tsv, etc.) For more detail, see: https://docs.google.com/document/d/1PmcVs7vg7Th-cGC-UrX8rAhKUHIzOI-uIOh69_mvdlw/edit; Pernet et al., 2019; bids.neuroimaging.io.

Many of these can be done using matlab tools to generate data (https://github.com/bids-standard/bids-starter-kit and https://github.com/sccn/bids-matlab-tools).

https://github.com/bids-standard/bids-matlab could be used to make analysis scripts more general.

Do reach out to the BIDS community and/or myself [Cyril Pernet] if you need help.

A classification of experiments could also be useful (using, say, the cognitive atlas ontology [https://www.cognitiveatlas.org/]).

2 Introduction

a) The introduction needs to better explain the SPN – how it is generated, what it is suppose to reflect/measure, how the potential is summarized in a single mean, etc.

b) Please add a paragraph at the end of the introduction telling readers more about the data, how they can get access to them, how they can reproduce your analyses, and what else it is possible to do with the data.

3 Figures

a) Related to the previous point, figures 1-3 are of limited usefulness to the reader. Please replace them with a single figure that includes a small number of illustrative SPN waves and a small number of illustrative topographic difference maps to clarify the explanation of the SPN (see previous point). Please include no more than four panels in reach row of the figure, and please ensure that the caption for the new figure 1 adequately explains all the panels in the figure.

b) The present figure 1 can then become Figure 1—figure supplement 1, and likewise for the present figures 2 and 3.

c) Please consider grouping the panels within each of these three figure supplements according to, say, a cognitive atlas ontology [https://www.cognitiveatlas.org/] (or similar).

d) Also, for the difference maps, please consider avoiding jet and using instead a linear luminance scale (eg, https://github.com/CPernet/brain_colours).

e) Figure 4b. Please delete the bar graph in the inset and give the three mean values (and confidence intervals) in the figure caption.

f) Figure 6. Please delete the bar graph (panel C) and give the four mean values (and confidence intervals) in the figure caption.

4 More complete reporting

a) Please include a table (as a Supplementary file) that provides information about the electrodes and time windows used in each of the studies. If the electrodes and/or time windows varied across the SPN waves within a study, please explain why.

b) Along similar lines, what stimulus types and tasks were used for what fraction of the SPNs in the dataset? SPNs can be measured for many types of regularity and varies with both stimulus parameters and task (as noted by the authors). This means that the SPN varies across studies not just by neural noise and measurement noise, but also by differences in neural signal. This is not necessarily a weakness of the analyses reported here, but it should be noted, so please include a table (as a Supplementary file) that provides information about the stimulus types and tasks used in the studies.

5 Further analyses

a) The Electrode Choice analysis estimates the effect of a priori selection of electrode ROIs, by comparing three such ROIs, each a smaller subset of the other. Perhaps not surprisingly the responses are highly correlated, and the overall effects change little. I would recommend considering more data-driven approaches to spatial filtering and dimensionality reduction. There are many such approaches, but two that come to mind are reliable components analysis (Dmochowski et al., 2012; Dmochowski et al., 2014; Dmochowski and Norcia, 2015) and non-parametric spatiotemporal clustering (Maris and Oostenveld, 2007). It would be interesting to see if approaches such as these would produce different effect sizes or identified a different set of electrodes across the many studies in the dataset. This could also serve as an assessment of the ability of these methods to identify the same spatial filters / clusters of electrodes across a large dataset. Given that these methods are widely used in the EEG/MEG-community, this would considerably enhance the impact of the manuscript. The spatiotemporal approach could also help address the time window selection issue, as outlined in my next point.

b) The authors acknowledge that the selection of time window is an important parameter that can influence the results, but do not include any analysis of this. Please include an analysis of the effect of time window selection across the many datasets. Also, please comment on how it is possible to measure and separate early and late effects using event-related EEG.

c) The analysis of the effect of pre-processing is also of potential broad interest to researchers doing EEG, but it is limited in impact, because only one study is re-analyzed. I understand that it may be impractical to re-analyze every study in the dataset 12 different ways, but re-analyzing a single dataset seems to have limited utility in the context of the current manuscript. Please either extend this analysis or remove it from the manuscript.

d) Re pre-processing pipelines, please consider discussing some of the recommendations in Pernet et al. [doi.org/10.1038/s41593-020-00709-0]

6 Limitations associated with using data from just one group.

The overall approach to the cumulative research program and the specific suggestions to reduce the experimental bias and improve researchers' self-awareness is indeed helpful but in line with the previously known findings on reproducibility limitations. Furthermore, a re-analysis, which specifically relies on single site single group acquisitions, poses additional issues or specific bias (e.g., data acquisition, methodology, experimental design, data pre-processing, etc.) to the generalizability of the results that the authors should have considered. A comparison with an independent sample of data would make the analysis less autoreferential and methodologically stronger. If possible, please include a comparison with an independent sample of data: if such a comparison is not possible, please discuss the limitations and potential biases associated with using data from just one group.

7 Awarding grades

The authors grading their work does not bring anything to our understanding of reproducibility; please remove.

8 Advice for other researchers

The manuscript currently has a strong focus on SPN research and in places almost reads like a 10-year report on scientific practice within a specific research group. This is undoubtedly valuable, but of limited general interest. So, I hope the authors will forgive an open-ended suggestion. Namely that they carefully consider the ways in which their findings might be valuable to researchers outside their own sphere of interest. I have suggested a few ways above, but perhaps the authors can think of others. What recommendations would the authors make to scientists working in other domains of brain imaging, based on the findings?

9 Publication bias

a) Line 145. "A meta-analysis based on the published SPNs along would overestimate...". It is essential that you perform such a meta-analysis. What you have done is compare the means computed from all vs. published, but not set-up a meta-analysis in which the mean is estimated from the means and variances of published studies alone (I suggest dmetar which is super easy to use even if you never used R / R-studio before; https://bookdown.org/MathiasHarrer/Doing_Meta_Analysis_in_R/). That extra analysis will be very useful because, people do not have access to the data to compute the mean and thus use this meta-analytical approach (how much this will differ from -1.17uV?; in theory it should bring the results closer to -0.94uV)

10 Low statistical power

– Power analysis: You need to explain better the analysis. It is also essential that you demonstrate that a 2nd order polynomial fits better than a 1st order using AIC and RMSE. A second aspect is that you cannot predict effect size from amplitude because the polynome you have was obtained for the entire dataset. This is a typical machine learning regression setting: you need to cross-validate to produce an equation than generalizes -- given the goal is to generalize across experiments, a natural cross validation scheme would be the experimental belonging (fit all data but one experiment, test – iterate to optimize parameters).

– Nonlinearity: to further look into that question you need the complementary plot Cohens'd vs. variance (or 3d plot d, variance, amplitude but sometimes it's hard to see). Given that d = mean/std, the nonlinearity arises because there is a 'spot' where variance behave strangely (likely for the midrange amplitude values). This also relates to your discussion on the number of trials because the number of trials increases precision and in principle reduce between subject variance; depending on expected amplitude the number of trials should therefore be adjusted to reduce variance bias.

11 Miscellaneous

a) Please provide links to the code for the analysis presented in the paper.

b) Please consider making the data-viewing app agnostic wrt operating system.

c) Re the headline effects reported in figure 5: the explanation of these effects in the in the repository needs to be more detailed.

d) Line 148. Please say more about the reasons why 68 SPNs "will likely in the file drawer forever".

e) Line 191 onwards. When talking about pilot studies please add references (for instance Lakens (doi.org/10.1016/j.jesp.2017.09.004). Also, please consider discussing the smallest effect size of interest instead of absolute effect size.

---

## [Author Response]

Essential Revisions1 Data sharingWe encourage the authors to make their data available in a way that is FAIR. As it stands the data are Findable and Accessible, but not much Interoperable or Reusable. For example, using BIDS elecrophysiological derivatives and given the authors experience with EEGLAB, they could use the.set format along with a flat structure and proper metadata (particiants.tsv, data_description.json, channels.tsv, electrodes.tsv, etc.) For more detail, see: https://docs.google.com/document/d/1PmcVs7vg7Th-cGC-UrX8rAhKUHIzOI-uIOh69_mvdlw/edit; Pernet et al., 2019; bids.neuroimaging.io.Many of these can be done using matlab tools to generate data (https://github.com/bids-standard/bids-starter-kit and https://github.com/sccn/bids-matlab-tools).https://github.com/bids-standard/bids-matlab could be used to make analysis scripts more general.Do reach out to the BIDS community and/or myself [Cyril Pernet] if you need help.A classification of experiments could also be useful (using, say, the cognitive atlas ontology [https://www.cognitiveatlas.org/]).

We have expanded our catalogue and supporting materials, and made the data as Finable, Accessible, Interoperable and Reusable (FAIR) as possible. As recommended, we have produced BIDS files for each of the 40 projects and included raw data as well as processed data (see Complete Catalogue-BIDS format, https://osf.io/e8r95/). We have now explained this in a new section after the introduction entitled “The complete Liverpool SPN catalogue”. One part reads:

“The SPN catalogue on OSF (https://osf.io/2sncj/) is designed to meet the FAIR standards (Findable, Accessible, Interoperable and Reusable). […] They can also run alternative analyses that depart from the original pipeline at any given stage.”

With all available raw data, the catalogue is now over 900 GB (up from the original 5 GB). This massively increases future possible uses of the catalogue. For instance, one could reanalyse the raw data using an alternative pipeline or run Time-Frequency analysis on cleaned data.

2 Introductiona) The introduction needs to better explain the SPN – how it is generated, what it is suppose to reflect/measure, how the potential is summarized in a single mean, etc.

We had already included some basics on the SPN and SPN catalogue in BOX 1. We have now provided more in the main manuscript. Furthermore, we have produced an ‘SPN gallery’, with one page for each of the 249 SPNs (This is available on OSF in the SPN analysis and guidebooks folder). The SPN gallery provides all information for anyone who wishes to see how a particular SPN was processed. The new section explains:

“On open science framework (https://osf.io/2sncj/) we provide several supplementary files, including one called ‘SPN gallery’ (see SPN analysis and guidebooks folder). […] Key mapping was usually shown only on the response screen, to avoid anticipation of a motor response.”

b) Please add a paragraph at the end of the introduction telling readers more about the data, how they can get access to them, how they can reproduce your analyses, and what else it is possible to do with the data.

We have now clarified this in a new section (see point 1). We have also elaborated on other things that researchers can do with the data:

“Although this paper focuses on meta-science, we can briefly summarize the scientific utility of the catalogue. […] The SPN catalogue also allows meta-analysis of other ERPs, such as P1 or N1, which may be systematically influenced by stimulus properties (although apparently not W-load).”

3 Figuresa) Related to the previous point, figures 1-3 are of limited usefulness to the reader. Please replace them with a single figure that includes a small number of illustrative SPN waves and a small number of illustrative topographic difference maps to clarify the explanation of the SPN (see previous point). Please include no more than four panels in reach row of the figure, and please ensure that the caption for the new figure 1 adequately explains all the panels in the figure.b) The present figure 1 can then become Figure 1—figure supplement 1, and likewise for the present figures 2 and 3.c) Please consider grouping the panels within each of these three figure supplements according to, say, a cognitive atlas ontology [https://www.cognitiveatlas.org/] (or similar).

We agree that these figures were of limited use, and we have replaced them with one representative Figure 1. Importantly, Figure 1 is a sample from the ‘SPN gallery’ on OSF, which has one such page for each of the 249 SPNs. The SPN gallery gives all information about stimuli and details of each experiment – and is thus far more informative than the original illustrative figures in the paper.

d) Also, for the difference maps, please consider avoiding jet and using instead a linear luminance scale (eg, https://github.com/CPernet/brain_colours).

Using the code provided, we obtained the luminance corrected scale and applied it to all topoplots in the SPN gallery and in the manuscript.

e) Figure 4b. Please delete the bar graph in the inset and give the three mean values (and confidence intervals) in the figure caption.

Figure 4 is now Figure 2 in the revised manuscript. We have deleted the bar graph and added a funnel plot of results from the new meta-analysis. The three mean values are now incorporated in text about meta-analysis. We feel these would not work in the caption now that the section has been expanded (see point 9 for new section on meta-analysis with Figure and caption).

f) Figure 6. Please delete the bar graph (panel C) and give the four mean values (and confidence intervals) in the figure caption.

Figure 6 is now Figure 4 in the revised manuscript (note also revised topoplot):

Again, we feel the text is a better place for the means than the Figure caption:

“Average SPN amplitude was -0.98 microvolts [95%CI = -0.88 to -1.08] with the original cluster, -0.961 [-0.87, -1.05] microvolts for cluster 1, -1.12 [-1.01, -1.22] microvolts for cluster 2, -0.61 [-0.55, -0.66] microvolts for cluster 3.”

4 More complete reportinga) Please include a table (as a Supplementary file) that provides information about the electrodes and time windows used in each of the studies. If the electrodes and/or time windows varied across the SPN waves within a study, please explain why.

On OSF we had already included a supplementary database of all electrodes and time windows used in each study (SPN effect size and power.xlsx). This is now referenced more prominently (see also point 1):

“We have enhanced reusability with several Supplementary files on OSF (see SPN analysis and guidebooks folder). These give all the technical details required for reproducible EEG research, as provided by OHBM COBIDAS MEEG committee in the Organization for Human Brain Mapping (Pernet et al., 2020). For instance, the file ‘SPN effect size and power V8.xlsx’ has one worksheet for each project. This file documents all extracted ERP data along with details about the electrodes, time windows, ICA components removed, and trials removed.”

The new SPN gallery also provides more information about electrodes and time windows, along with uniform visualizations of all 249 SPNs.

Electrodes and time windows did vary between studies, as discussed at length in the p-hacking section. However, they did not vary between SPN waves within an experiment. We do not think it would be feasible to work through all between-experiment inconsistencies on a case-by-case basis in the manuscript.

b) Along similar lines, what stimulus types and tasks were used for what fraction of the SPNs in the dataset? SPNs can be measured for many types of regularity and varies with both stimulus parameters and task (as noted by the authors). This means that the SPN varies across studies not just by neural noise and measurement noise, but also by differences in neural signal. This is not necessarily a weakness of the analyses reported here, but it should be noted, so please include a table (as a Supplementary file) that provides information about the stimulus types and tasks used in the studies.

The new SPN gallery provides all details about stimuli and tasks used for each of the 249 SPNs. We have also added this summary in the manuscript:

“On open science framework (https://osf.io/2sncj/) we provide several supplementary files, including one called ‘SPN gallery’ (see SPN analysis and guidebooks folder). […] Key mapping was usually shown only on the response screen, to avoid anticipation of a motor response.”

We now elaborate the point about signal and noise in the section on publication bias:

“In other words, the SPN varies across studies not only due to neural noise and measurement noise, but also due to experimental manipulations affecting the neural signal (although W-load and Task were similar in unpublished work).”

Finally, the new meta-analysis covers this point about study heterogeneity much more explicitly (see point 9).

5 Further analysesa) The Electrode Choice analysis estimates the effect of a priori selection of electrode ROIs, by comparing three such ROIs, each a smaller subset of the other. Perhaps not surprisingly the responses are highly correlated, and the overall effects change little. I would recommend considering more data-driven approaches to spatial filtering and dimensionality reduction. There are many such approaches, but two that come to mind are reliable components analysis (Dmochowski et al., 2012; Dmochowski et al., 2014; Dmochowski and Norcia, 2015) and non-parametric spatiotemporal clustering (Maris and Oostenveld, 2007). It would be interesting to see if approaches such as these would produce different effect sizes or identified a different set of electrodes across the many studies in the dataset. This could also serve as an assessment of the ability of these methods to identify the same spatial filters / clusters of electrodes across a large dataset. Given that these methods are widely used in the EEG/MEG-community, this would considerably enhance the impact of the manuscript. The spatiotemporal approach could also help address the time window selection issue, as outlined in my next point.b) The authors acknowledge that the selection of time window is an important parameter that can influence the results, but do not include any analysis of this. Please include an analysis of the effect of time window selection across the many datasets. Also, please comment on how it is possible to measure and separate early and late effects using event-related EEG.

A and B are interesting points, and we treat them together. One problem with our original analysis in the P-hacking section was that it conflated space and time – inter-cluster correlations were less than 1, but we could not determine the extent to which variability in electrode cluster and time window were responsible for the imperfect correlation. We have now removed this problem by redoing our analysis of a priori electrode clusters on consistent time windows (results are in what is now Figure 4).

Following this, our new spatio-temporal clustering analysis addresses issues with space and time together. Crucially, we found that this data-driven approach homed in on the same clusters and time windows as our a priori posterior electrode choice. There were no surprising symmetry related responses at frontal electrodes or at unexpected time points. The new section in the manuscript reads:

“Spatio-temporal clustering

The above analysis examines consequences of choosing different electrode clusters a priori, while holding time window constant. […] There was a strong correlation between this and SPN from each a priori cluster (Pearson’s r ranged from 0.719 to 0.746, p <.001, Figure 7).”

c) The analysis of the effect of pre-processing is also of potential broad interest to researchers doing EEG, but it is limited in impact, because only one study is re-analyzed. I understand that it may be impractical to re-analyze every study in the dataset 12 different ways, but re-analyzing a single dataset seems to have limited utility in the context of the current manuscript. Please either extend this analysis or remove it from the manuscript.

We agree this section was of limited value in the previous version. We elected to replace it with something far more ambitious rather than removing it. After all, pipeline comparison is a pertinent topic for all neuroscience researchers. We thus reanalysed all of the 5 experiments (from project 13) using BESA, and our familiar EEGLAB and Matlab. In BESA there was no ICA cleaning, and low pass filters were different. As we can see in the new Figure 8, the waves are similar. This increases confidence that our SPNs are not overly altered by conventions in the pre-processing pipeline. The following paragraph has been added to the manuscript to describe the two different pipelines:

“Pre-processing pipelines

There are many pre-processing stages in EEG analysis (Pernet et al., 2020). For example, our data is re-referenced to a scalp average, low pass filtered at 25Hz, downsampled to 128 Hz and baseline corrected using a -200 to 0 ms pre-stimulus interval. [..] Earlier analysis stages are less susceptible to P Hacking because changing them does not predictably increase or decrease the desired ERP effect.”

In sum, we can demonstrate convergence between results obtained with different pipelines. This was not a foregone conclusion: Neuroimaging results can vary greatly when pipelines and software are used on the same raw data. We have now made this point in the manuscript:

“Reassuringly, we found most effects could be recreated with a variety of justifiable analysis pipelines. Data-driven and a priori approaches gave similar results, and different EEGLAB and BESA pipelines gave similar results. This was not a foregone conclusion – related analysis in fMRI has revealed troubling inconsistencies (Lindquist, 2020). It is advisable that all researchers compare multiple analysis pipelines to assess vibration of effects and calibrate their confidence accordingly.”

d) Re pre-processing pipelines, please consider discussing some of the recommendations in Pernet et al. [doi.org/10.1038/s41593-020-00709-0]

We have now cited this paper as an authoritative source of advice EEG pipelines:

“We have enhanced reusability with several Supplementary files on OSF (see SPN analysis and guidebooks folder). These give all the technical details required for reproducible EEG research, as provided by OHBM COBIDAS MEEG committee in the Organization for Human Brain Mapping (Pernet et al., 2020). For instance, the file ‘SPN effect size and power V8.xlsx’ has one worksheet for each project. This file documents all extracted ERP data along with details about the electrodes, time windows, ICA components removed, and trials removed.”

More significantly, we have followed some these recommendations when expanding the catalogue and upgrading supplementary material. For instance, we have added 95% CI around all SPNs in the SPN gallery and in Box 1.

6 Limitations associated with using data from just one group.The overall approach to the cumulative research program and the specific suggestions to reduce the experimental bias and improve researchers' self-awareness is indeed helpful but in line with the previously known findings on reproducibility limitations. Furthermore, a re-analysis, which specifically relies on single site single group acquisitions, poses additional issues or specific bias (e.g., data acquisition, methodology, experimental design, data pre-processing, etc.) to the generalizability of the results that the authors should have considered. A comparison with an independent sample of data would make the analysis less autoreferential and methodologically stronger. If possible, please include a comparison with an independent sample of data: if such a comparison is not possible, please discuss the limitations and potential biases associated with using data from just one group.

We have now mentioned this limitation in the new section on the SPN catalogue:

“SPNs from other labs are not yet included in the catalogue (Höfel and Jacobsen, 2007a, 2007b; Jacobsen, Klein, and Löw, 2018; Martinovic, Jennings, Makin, Bertamini, and Angelescu, 2018; Wright, Mitchell, Dering, and Gheorghiu, 2018). Steady state visual evoked potential responses to symmetry are also unavailable (Kohler, Clarke, Yakovleva, Liu, and Norcia, 2016; Kohler and Clarke, 2021; Norcia, Candy, Pettet, Vildavski, and Tyler, 2002; Oka, Victor, Conte, and Yanagida, 2007). However, the intention is to keep the catalogue open, and the design allows many contributions. In the future we hope to integrate data from other labs. This will increase the generalizability of our conclusions.”

And in the discussion

“The focus on a single lab inevitably introduces some biases and limitations. Other labs may use different EEG apparatus with more channels. This would inevitably have some effect on the recordings. More subtle differences may also matter: For instance, other labs may use more practice trials or put more stress on the importance of blink suppression. However, the heterogeneity of studies and pipelines explored here ensures reasonable generalizability. We are confident that our conclusions are relevant for SPN researchers with different apparatus and conventions.”

7 Awarding gradesThe authors grading their work does not bring anything to our understanding of reproducibility; please remove.

We have removed this and instead included more advice for other researchers, as described next.

8 Advice for other researchersThe manuscript currently has a strong focus on SPN research and in places almost reads like a 10-year report on scientific practice within a specific research group. This is undoubtedly valuable, but of limited general interest. So, I hope the authors will forgive an open-ended suggestion. Namely that they carefully consider the ways in which their findings might be valuable to researchers outside their own sphere of interest. I have suggested a few ways above, but perhaps the authors can think of others. What recommendations would the authors make to scientists working in other domains of brain imaging, based on the findings?

We feel this is perhaps the most substantial criticism. We have addressed it in two ways. First, at the end of each section, we have replaced the auto-referential grades with a one paragraph recommendation for other researchers.

At the end of Publication bias:

“Some theoretically important effects can appear robust in meta-analysis of published studies, but then disappear once the file drawer studies are incorporated. Fortunately, this does not apply to the SPN. We suggest that assessment of publication bias is a feasible first step for other researchers undertaking a catalogue-evaluate exercise.”

At the end of Low statistical Power, we went beyond the original article by comparing sample sizes in may published ERP papers and drawing more general conclusions:

“When planning our experiments, we have often assumed that 24 is a typical sample size in EEG research. […] Larger samples reduce type II error, give more accurate estimates of true effect size, and facilitate additional exploratory analyses on the same data sets.”

At the end of P-hacking

“It is easier to publish a simple story with a beautiful procession of significant effects. […] It is advisable that all researchers compare multiple analysis pipelines to assesvibration of effects and calibrate their confidence accordingly”

At the end of HARKing

“Our recommendation is to use pre-registration to combat HARKing. Perhaps unregistered EEG experiments will be considered unpublishable in the not-too-distant future.”

These four recommendations are locally important. Our one overarching recommendation is simply for other researchers to do what we have done: They should compile complete public catalogues and then use them to evaluate their own data and practices using Bishop’s four horsemen framework. In informal terms, we want readers to think “Good question – am I delivering meaningful results or not? This is rather important, and nobody else is going to check for me, even if I make all my data sets publicly available. Maybe I should also do a big catalogue-evaluate exercise like this”. We previously left this overarching recommendation implicit because actions speak louder than words, and there are already many authoritative meta-science commentaries urging improvement. However, we have now added one more advice paragraph Discussion section:

“While some readers may not be interested in the validity of SPN research per se, they may be interested in this meta-scientific exercise. Our one overarching recommendation is that researchers could consider conducting comparable catalogue – evaluate exercises on many phenomena. In fact, this may be essential for cumulative science. It has been argued that research should be auditable (Nelson et al., 2018), but Researcher A will rarely be motivated to audit a repository uploaded by Researcher B, even if the datasets are FAIR. To fight the replication crisis, we must actively look in the mirror, not just passively let others snoop through the window.”

9 Publication biasa) Line 145. "A meta-analysis based on the published SPNs along would overestimate...". It is essential that you perform such a meta-analysis. What you have done is compare the means computed from all vs. published, but not set-up a meta-analysis in which the mean is estimated from the means and variances of published studies alone (I suggest dmetar which is super easy to use even if you never used R / R-studio before; https://bookdown.org/MathiasHarrer/Doing_Meta_Analysis_in_R/). That extra analysis will be very useful because, people do not have access to the data to compute the mean and thus use this meta-analytical approach (how much this will differ from -1.17uV?; in theory it should bring the results closer to -0.94uV)

We have conducted the meta-analysis and added funnel plots to Figure 2. Weighted mean SPN amplitude was found to be -0.954 μV. The funnel plots showed some asymmetry, but this cannot be the result of publication bias because it was still present when all SPN in the file drawer were incorporated. We also found it instructive to meta-analyse P1 and N1 peaks, because these signals are not likely to influence publication success in the same way. The publication bias section now includes details of this meta-analysis:

“To further explore these effects, we ran three random-effects meta-analyses (using metamean function from the dmetar library in R). The first was limited to the 115 published SPNs (as if they were the only ones available). Weighted mean SPN amplitude was -1.138 microvolts [95%CI = -1.290 to -0.986]. These studies could be split in groups, as they use different types of stimuli and tasks, but here we focus on the overall picture.”

The funnel plot in Figure 2A shows published SPNs as green data points. These are not symmetrically tapered: Less accurate measures near the base of the funnel are skewed leftwards. This is a textbook fingerprint of publication bias, and Egger’s test found that the asymmetry was significant (bias = -6.866, t (113) = -8.45, p <.001). However, a second meta-analysis, with all SPNs, found mean SPN amplitude was reduced, at -0.954 microvolts [-1.049; -0.860], but asymmetry in the funnel plot was still significant (bias = -4.546, t (247) = -8.20, p <.001). Furthermore, the asymmetry was significant in a third meta-analysis on the unpublished SPNs only (Weighted mean SPN = -0.801 microvolts [-0.914; -0.689], bias = -2.624, t (132) = -3.51, p <.001).

P1 peak and N1 trough amplitudes from the same trials provide an instructive comparison. P1 peak was measured as maximum amplitude achieved between 100 and 200 ms post stimulus onset. The N1 trough was measured as the minimum between 150 and 250 ms, as a difference from P1 peak (see inset ERP waveforms Figure 2). Our papers do not need large P1 and N1 components, so these are unlikely to have a systematic effect on publication. P1 peak was essentially identical in published and unpublished work (4.672 vs. 4.686, t (195.11) = 0.067, p = .946, Figure 2B). This is potentially an interesting counterpoint to the SPN. However, the N1 trough was larger in published work (-8.736 vs. -7.155. t (183.61) = -5.636, p <.001, Figure 2C). Funnel asymmetry was also significant for P1 (bias = 5.281, t (247) = 11.13, p< 0.001) and N1 (bias = -5.765, t (247) = -13.15, p <.001).

We conclude that publication bias does not explain the observed funnel plot asymmetry. It could be that larger brain responses are inherently more variable between trials and participants. This could cause funnel asymmetry which may be erroneously blamed on publication bias. Furthermore, larger sample sizes may have been selected based on the expectation of a small effect size (see Zwetsloot et al., 2017 for detailed analysis of funnel asymmetry).

This analysis was based on publication status in October 2020. Four more studies have since escaped the file drawer and joined the literature. By January 2022, published and unpublished SPNs had similar samples (26.17 vs 26.81, p = .596). However, recategorizing these 4 studies does not dramatically alter the results (File drawer weighted mean SPN = -0.697 [-0.818; -0.577], funnel asymmetry = -2.693; t (98) = -3.23, p = .002); Literature weighted mean SPN = – 1.127 [-1.256; -0.997] funnel asymmetry = -5.910, t (147) = -8.66, (p <.001). Finally, one can also take a different approach using 84 SPNs from 84 independent groups. Results were very similar, as described in Supplementary independent groups analysis (available in the SPN analysis and guidebooks folder on OSF).

10 Low statistical power– Power analysis: You need to explain better the analysis. It is also essential that you demonstrate that a 2nd order polynomial fits better than a 1st order using AIC and RMSE. A second aspect is that you cannot predict effect size from amplitude because the polynome you have was obtained for the entire dataset. This is a typical machine learning regression setting: you need to cross-validate to produce an equation than generalizes -- given the goal is to generalize across experiments, a natural cross validation scheme would be the experimental belonging (fit all data but one experiment, test – iterate to optimize parameters).– Nonlinearity: to further look into that question you need the complementary plot Cohens'd vs. variance (or 3d plot d, variance, amplitude but sometimes it's hard to see). Given that d = mean/std, the nonlinearity arises because there is a 'spot' where variance behave strangely (likely for the midrange amplitude values). This also relates to your discussion on the number of trials because the number of trials increases precision and in principle reduce between subject variance; depending on expected amplitude the number of trials should therefore be adjusted to reduce variance bias.

These are very important points. We have included a new supplementary document on OSF that covers these topics. In the manuscript we link to it with this new sentence:

“The curve tails off for strong SPNs, resulting in a nonlinear relationship. The second order polynomial trendline was a better fit than the first order linear trendline (see supplementary polynomial regression in the SPN analysis and guidebooks folder on OSF).”

The new supplementary polynomial regression analysis is based on the above recommendations.

11 Miscellaneousa) Please provide links to the code for the analysis presented in the paper.

On OSF we now have a folder explicitly dedicated to the analysis used in this paper (see compressed ‘Analysis in *eLife* paper’ in the SPN apps and guidebooks folder, https://osf.io/2sncj/). This link is now included in the manuscript.

b) Please consider making the data-viewing app agnostic wrt operating system.

Done

c) Re the headline effects reported in figure 5: the explanation of these effects in the in the repository needs to be more detailed.

We have taken this part of the figure out because it may be a case of dwelling on SPN-specific issues. In the repository we have now added a DOI and a longer description of each effect (see Famous ANOVA effect sizes.xlsx > sheet2).

d) Line 148. Please say more about the reasons why 68 SPNs "will likely in the file drawer forever".

As part of our attempts to make the paper less self-referential we removed this future publication bias section, which was a bit speculative anyway. A common reason some projects will be in the file drawer forever is because other more conclusive versions end up being published instead.

e) Line 191 onwards. When talking about pilot studies please add references (for instance Lakens (doi.org/10.1016/j.jesp.2017.09.004). Also, please consider discussing the smallest effect size of interest instead of absolute effect size.

We have now cited this paper when making the point about the low value of pilot studies for estimating effect size and power.

“It is less well-known that one pilot experiment does not provide a reliable estimate of effect size (especially when the pilot itself has a small sample; Albers and Lakens, 2018).”

We have also added this about smallest effect size:

“Alternatively, researchers may require a sample that allows them to find the minimum effect that would still be of theoretical interest. Brysbaert (2019) suggests this may often be ~0.4 in experimental psychology, and this requires 52 participants. Indeed, the theoretically interesting effect of Task on SPN amplitude could be in this range.”